# UNISOLVER: PDE-CONDITIONAL TRANSFORMERS ARE UNIVERSAL NEURAL PDE SOLVERS

## ABSTRACT

Deep models have recently emerged as a promising tool to solve partial differential equations (PDEs), known as neural PDE solvers. While neural solvers trained from either simulation data or physics-informed loss can solve PDEs reasonably well, they are mainly restricted to a few instances of PDEs, e.g. a certain equation with a limited set of coefficients. This limits the generalization of neural solvers to diverse PDEs, impeding them from being practical surrogate models for numerical solvers. In this paper, we present the Universal Neural PDE Solver (Unisolver) capable of solving a wide scope of PDEs by training a novel Transformer model on diverse data and conditioned on diverse PDEs. Instead of purely scaling up data and parameters, Unisolver stems from the theoretical analysis of the PDE-solving process. Inspired by the mathematical structure of PDEs, that a PDE solution is fundamentally governed by a series of PDE components, such as equation symbols, coefficients, and boundary conditions, we define a complete set of PDE components and flexibly embed them as domain-wise (e.g. equation symbols) and point-wise (e.g. boundaries) deep conditions for Transformer PDE solvers. Integrating physical insights with recent Transformer advances, Unisolver achieves consistent state-of-the-art results on three challenging large-scale benchmarks, showing impressive performance gains and favorable PDE generalizability.

## 1 INTRODUCTION

Partial differential equations (PDEs) are essential for numerous scientific and engineering problems (Evans, 2022; Arnol'd, 2013), such as meteorology, electromagnetism and thermodynamics (Wang et al., 2023). Since it is usually hard to obtain an analytic solution for a PDE, numerical methods are widely explored (Ames, 2014). However, these numerical methods often require huge computation costs to generate a precise solution for each PDE. Recently, deep learning models have facilitated significant advancements across a wide range of domains (Devlin et al., 2019; Liu et al., 2021; Jumper et al., 2021) and have been applied to solving PDEs, i.e. neural PDE solvers (Karniadakis et al., 2021). Owing to their excellent capability to approximate nonlinear mappings, deep models can learn to fit pre-collected data (Li et al., 2021a) or physics-informed loss function (Raissi et al., 2019) and generalize in a flash to new samples, providing an efficient approach to solving PDEs.

As shown in Figure 1, previous neural solvers can be broadly categorized into two paradigms: physics-informed neural networks (PINNs) (Raissi et al., 2019) and neural operators (Li et al., 2021a). The former trains deep models using a formalized PDE loss function, while the latter solely relies on pre-collected data. However, for PINNs, while formulating the PDE equations as objective functions can ensure relatively accurate solutions, they struggle to generalize to new scenarios, necessitating retraining the model for each new task. Neural operators, on the other hand, directly learn from the data and tend to generalize better to diverse initial states and PDEs than PINNs. Nevertheless, purely based on training data may be insufficient to guide PDE solving. For example, in the case of a fluid governed by renowned Navier-Stokes equations, the typical task of neural operators is to predict future states based on past observations (Li et al., 2021a), while different viscosity coefficients and forcing terms will lead to distinct solutions even when the initial states stay the same. Thus, due to the omission of PDE information, current neural operators are mainly trained and tested on a limited set of PDEs. Notably, as neural solvers are expected to be efficient surrogate models of classical numerical solvers, generalization to various PDEs is essential for a practical neural solver.

Figure 1: Neural PDE solvers typically consist of two paradigms: physics informed and data driven. Our proposed Unisolver combines data-driven methods with physical insights from complete PDE components in a conditional modeling framework, thereby boosting generalizability and scalability.

To tackle the generalization deficiency, several works have been proposed by incorporating the PDE information into deep models or training models with a large-scale dataset. For example, message-passing neural PDE solver (Brandstetter et al., 2022) concatenates the PDE coefficients with inputs. PDEformer (Ye et al., 2024) formalizes the PDE equation as a computation graph and employs the graph Transformer (Ying et al., 2021) to aggregate PDE information. Although these methods have explored the potential of training models with both data and PDE information, they fail to consider a complete set of PDE information, thereby limiting their generalizability in some aspects. As for the other branches, such as DPOT (Hao et al., 2024), they purely scale up the training sets with diverse PDEs and expect the generalizability emerges from large data and parameters. Although models can implicitly extract PDE information from observations, the extraction process is inherently complex and resembles the challenges associated with solving inverse problems (Karniadakis et al., 2021). Therefore, these models often end up fitting an insufficient or vague representation of the underlying observation distribution, which ultimately hampers their generalizability to broader PDE solving.

Going beyond prior methods, as shown in Figure 1, this paper presents Unisolver as a Universal Neural PDE solver. Concretely, Unisolver takes the advantages from both data-driven and physics-informed paradigms and empowers Transformer with favorable generalizability by introducing complete physics information as conditions. Instead of simply scaling up data and parameters, we are motivated from the theoretical analysis of PDE solving and propose a complete set of PDE components. Further, drawing inspiration from the mathematical structure of PDEs, we propose to classify PDE components into domain-wise and point-wise categories according to their effect on the final solution and aggregate them as two types of deep PDE conditions. Afterward, to capture the special influence of different condition types on the hidden representations of inputs, we separate the hidden space into two subspaces and integrate these deep PDE conditions into the hidden representations of inputs in a decoupled way. We conduct extensive experiments on our own generated dataset and two large-scale benchmarks with various PDE components, where Unisolver achieves consistent state-of-the-art with sharp relative gains. Overall, our contributions are summarized as follows:

- We introduce Unisolver as a conditional Transformer architecture utilizing the embedded PDE information completely, marking the first demonstration of the potential of the canonical Transformer as a scalable backbone for solving multitudinous PDEs universally.

- Motivated by the mathematical structure of PDEs, we define the concept of complete PDE components, classify them into domain-wise and point-wise categories, and derive a decoupled conditioning mechanism for introducing physics information into PDE solving.

- Unisolver achieves consistent state-of-the-art performances across three large-scale benchmarks with impressive relative gains and presents favorable generalizability and scalability.

## 2 RELATED WORK

### 2.1 NEURAL PDE SOLVERS

Previous neural PDE solvers can be roughly categorized into the following two paradigms (Wu et al., 2024). The first paradigm is physics-informed neural networks (PINNs) (Raissi et al., 2019), which optimize the deep model by formalizing the PDE equations as objective functions. During training,

the model outputs and gradients will gradually satisfy the targeted PDE, thereby successfully instantiating the solution as a deep model. However, it is usually hard for PINNs to generalize to unseen PDEs, limiting their applications in broader practice (Wang et al., 2023). Another booming direction is neural operators, which learn from extensive data to approximate functional dependence between input and output Banach spaces (Lu et al., 2021; Kovachki et al., 2023). Among various neural operators, FNO (Li et al., 2021a) and its variants (Li et al., 2023c; Rahman et al., 2023; Wen et al., 2022) are popular and well-established. FNO (Li et al., 2021a) effectively approximates the kernel integral operator in the frequency domain through Fourier transformation. Besides, LSM (Wu et al., 2023) generalizes the classical spectral method in latent space to tackle the curse of dimensionality. Recently, given the impressive progress achieved by Transformers (Vaswani et al., 2017), they have also been applied to solve PDEs. Existing methods treat inputs as a sequence of tokens and adopt the attention mechanism to approximate integral for solving PDEs. OFormer (Li et al., 2023a) and GNOT (Hao et al., 2023) treat each mesh point as a token and utilize the linear Transformer to avert the complexity problem. Factformer (Li et al., 2023b) axially factorizes the attention block to boost the model efficiency. Recently, Transolver (Wu et al., 2024) proposes to learn the intrinsic physical states as tokens behind input meshes, deriving a physics-attention mechanism. Despite the success of neural operators, they are only tested on the dataset containing limited PDEs. The effectiveness of these methods under large datasets containing various PDEs has not been fully explored.

## 2.2 GENERALIZABLE PDE SOLVERS

In addition to model architectures, the generalizability of neural solvers, the major advantage of numerical solvers, has also been explored. The research mainly lies in the following two directions.

**Incorporating PDE information** To guide the PDE-solving process, PDE information has been explored in many deep models. For example, PINO (Li et al., 2021b) imposes explicit equation constraints at a higher resolution to assist the learning of neural operators. CAPE (Takamoto et al., 2023) directly embeds PDE coefficients to adapt neural solvers to unseen equation coefficients. PROSE (Liu et al., 2023) and PITT (Lorsung et al., 2024) tokenize PDEs and embed mathematical expressions, enabling the transformer backbone to become aware of the underlying physics. PDE-former (Ye et al., 2024) represents the symbolic form of equations as a graph and the numeric components as nodes to optimize the processing of complex interactions between symbolic and numeric information. However, all of these methods, while incorporating equation information, do not leverage the mathematical structure of PDEs for complete and categorized embedding or integrating the prior information of equation symbols within the context of natural language. In contrast, Unisolver leverages the capabilities of large language models (LLMs) (Touvron et al., 2023) to semantically embed the equation symbolic information and categorize the complete equation components based on mathematical insights, thereby facilitating the modeling of generalizable physical correlations.

**Large-scale training** As a vital cornerstone of deep learning (Brown et al., 2020; He et al., 2022), recent research has also started to explore the effectiveness of large-scale training in solving PDEs. Subramanian et al. examine the scaling capabilities and transfer learning behaviors of FNO on three time-independent PDE families. MPP (McCabe et al., 2023) proposes an auto-regressive strategy to train on a broad fluid mechanics-oriented benchmark. DPOT (Hao et al., 2024) enhances MPP with a denoising method and trains a Fourier Transformer on massive PDE data comprised of 12 datasets. PDEformer (Ye et al., 2024) focuses on a 1D time-dependent PDE family and pre-trains a graph transformer on 3M samples under various equation conditions. ICON (Yang et al., 2023) trains a single neural operator capable of performing in-context learning across a wide range of differential equations. However, most of the existing methods fall short in effectively and completely integrating PDE information. This will be well addressed by Unisolver in a natural and generalizable way.

## 3 UNISOLVER

To tackle the incapability in generalization behind neural PDE solvers, we deeply dive into the PDE-solving process and present Unisolver to model the intricate interactions between initial observations and complete equation components, leading to a novel PDE-conditional Transformer model.

**Problem setup** To achieve ideal generalizability, we focus on the task of *universal neural PDE solving*. Let $\mathcal{D} \subset \mathbb{R}^d$ be a bounded continuous set and $\mathcal{M} = \{x_1, \ldots, x_n\}$ be an $n$-point discretization of $\mathcal{D}$ recording the coordinates of each point. For each observation pair, assume we have initial condition observations $\mathbf{X}$ as input and target quantities $\mathbf{Y}$ as output on the mesh $\mathcal{M}$, with the

Figure 2: Overview of universal neural PDE solving, taking the 2D mixed PDEs in Sec 4.3 as an example. Our model is jointly trained on diverse PDEs with varied initial conditions and governing PDE components, aiming for direct generalization to unseen PDEs in downstream tasks. The "Robin boundary" in gray is a valid boundary type despite not included in the example dataset.

governing PDE components $\mathcal{C}_{\text{PDE}}$ (e.g. PDE symbols, coefficients, etc.) which may vary for each observation. The universal neural PDE solving task is to approximate the input-PDE-output map $G : (\mathbf{X}, \mathcal{M}, \mathcal{C}_{\text{PDE}}) \to \mathbf{Y}$ across a diverse training dataset and generalize in a flash to unseen PDEs.

## 3.1 COMPLETE PDE COMPONENTS

To enable complete modeling of the PDE, we attempt to incorporate the *complete PDE components*, i.e. all the underlying components that affect solutions, into neural solvers.

**A motivating example** Here, we clarify the concept in the context of deep learning by considering the classical *vibrating string equation* with fixed endpoints as a motivating example, which can be solved explicitly, as shown in (Evans, 2022). The analytical solution is provided in Appendix E.

$$\partial_{tt}u - a^2\partial_{xx}u = f(x,t), \qquad (x,t) \in (0,L) \times (0,T), \qquad (1a)$$
$$u(0,t) = 0, \ u(L,t) = 0, \qquad t \in (0,T), \qquad (1b)$$
$$u(x,0) = \phi(x), \ \partial_t u(x,0) = \psi(x), \qquad x \in [0,L]. \qquad (1c)$$

From the solving process of the above motivating example, we pinpoint that the PDE is solved through complex interactions between a series of equation components, as detailed in Table 1. These components are referred to as the *complete PDE components* and exhibit two key shared characteristics. Specifically, the coefficient $a$ exerts the same influence over the entire domain, while the impact of the force $f$ is imposed point-wisely. This distinction inspires us to classify these components into two categories, *domain-wise* and *point-wise*, which better capture the intricate interactions.

Table 1: A detailed categorization of complete PDE components with corresponding examples.

| Groups | Input | Domain-wise components | | | Point-wise components | | |
|---|---|---|---|---|---|---|---|
| Components | Initial condition | Equation symbols | Equation coefficient | Boundary condition type | External force | Domain geometry | Boundary value function |
| Example | Eq. (1c) | Eq. (1a) | $a$ | Robin | $f(x,t)$ | $[0,L] \times [0,T]$ | Eq. (1b) |

Moreover, we explain the classification of the other components shown in Table 1. The equation formulation is defined as a domain-wise component due to its consistency across all locations. Domain geometry is categorized as a point-wise component since it is usually recorded as a binary mask and each point's inclusion is determined individually. Boundary conditions are more complicated due to their diverse forms, e.g. periodic and Robin boundary conditions. As a result, we use two components to represent boundary conditions precisely: the boundary condition type, treated as a domain-wise component, and the boundary value function, considered as a point-wise component.

## 3.2 UNIVERSAL COMPONENTS EMBEDDING

As described in Section 3.1, PDE solutions are obtained by intricate interactions between initial conditions and complete equation components which can be grouped into two categories. In previous works (Brandstetter et al., 2022; Takamoto et al., 2023), these equation components are coarsely and incompletely included as conditions to modulate the input observations. In this paper, we will elaborate on how Unisolver finely and completely embeds all considered PDE components (Table 1) into deep PDE conditions based on our insights from the mathematical analysis.

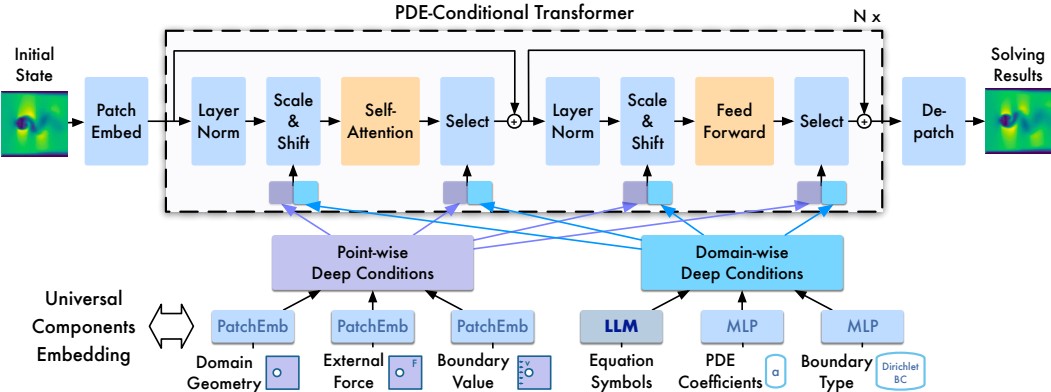

Figure 3: Overview of Unisolver. We universally embed all PDE components into deep conditions and employ a conditional Transformer to aggregate deep conditions in the decoupled subspace.

**Equation formulation** Since the mathematical symbols convey rich mathematical information, we utilize a Large Language Model (LLM) for symbolic embedding. Specifically, we adopt the recently released *LLaMA-3 8B* model [1] to embed the equation formulation. We attempt to leverage its understanding of prior mathematical information, which was learned from pre-training on 15 TB of language tokens, as well as its flexible encoding of unstructured PDE information. Technically, the input to the LLM is the LaTeX code of the equation. For example, the Eq. (1a) is prompted as

```
Prompt: "u_{tt} - a^2 u_{xx} = f(x,t)"
```

Then we take the output of the last Transformer block of the LLM and average representations along the sequence dimension, resulting in a 4096-dimensional embedding for each PDE. Notably, in the LLM embedding stage, we utilize mathematical symbols of the remaining equation components (e.g. coefficients and force terms) rather than their actual values in the prompt. For instance, we adopt symbol "a" in the above prompt rather than its concrete value to make the LLM focus on the key physics meaning of PDEs. The embedding of concrete values for the other components will be detailed in the next paragraph. After the LLM embedding stage, the hidden representations of PDE symbols are encoded by an MLP to align channel dimensions and obtain deep conditions.

**Other components** As we illustrated in Table 1, other components can be categorized as domain-wise and point-wise according to their effect on the final solution. Correspondingly, we adopt different embedding methods for these two types. For domain-wise components, including coefficients represented as real-valued vectors and boundary types similar to class labels, we embed them using two linear layers with an in-between SiLU activation function (Elfwing et al., 2018). Moreover, point-wise components like external force, binary geometry mask, and boundary value functions are essentially physical fields observed on mesh $\mathcal{M}$. We apply the same patchify embedding method used for input observations, transforming them into deep representation sequences.

**Deep condition consolidation** As shown in Figure 3, after universal components embedding, deep conditions within the same category are added together to consolidate their impact. This strategy prevents excessive separation of deep PDE conditions that could weaken the model's expressive capabilities, and thus will enhance representation learning for diverse PDE solving via joint training.

### 3.3 PDE-CONDITIONAL TRANSFORMER

We propose a conditional Transformer to adaptively fuse deep PDE conditions, embedded from the complete equation components, into hidden representations of inputs within decoupled subspaces.

**Subspace decoupling** We evenly split the hidden representations of the inputs along the channel dimension, with one half influenced by domain-wise deep conditions and the other half by point-wise deep conditions. Especially in multi-head attention (Vaswani et al., 2017), our proposed subspace decoupling is equivalent to assigning some heads to learn the impact of domain-wise conditions while others focusing on point-wise conditions. This leads to improved representation learning for both categories, and minimized interference between deep PDE conditions from two categories.

---
[1]https://ai.meta.com/blog/meta-llama-3/

**Deep condition aggregation**   We utilize MLPs to individually project domain-wise conditions and point-wise conditions into the corresponding subspace. After projection, domain-wise conditions are repeated along the sequence dimension to match the length of token sequence and ensure consistent physical guiding throughout the entire sequence. The transformed conditions convey both domain-wise and point-wise information, which are then integrated adaptively by aggregation functions.

As shown in Figure 3, we aggregate conditions either before or after the attention and feedforward modules within Transformer. Inspired by recent conditional Transformers like DiT (Peebles & Xie, 2023) and other conditional normalization approaches (Park et al., 2019; Perez et al., 2018), we take the aggregation paradigm to finely capture the intricate correlations between hidden inputs of initial observations and deep equation conditions. Specifically, we *scale* and *shift* the hidden representations of inputs based on the equation conditions. After passing through the Transformer modules, we use the equation conditions to softly *select* whether this information should be retained.

**Overall design**   Summarizing the above designs, we propose the Unisolver (Figure 3). Given input $\mathbf{X}$, it is projected to embeddings $\mathbf{X}^0$ using a patchify layer (Dosovitskiy et al., 2020). The complete PDE equation components $\mathcal{C}_{\text{PDE}}$ are embedded into deep conditions $\mathbf{C}_{\text{domain}}$ and $\mathbf{C}_{\text{point}}$ following Section 3.2. Suppose there are $N$ layers, the $n$-th layer of Unisolver can be formalized as:

$$
\begin{aligned}
\mathbf{I}_* &= \text{Concat}\left(\text{MLP}_*(\mathbf{C}_{\text{domain}}).\texttt{repeat}, \text{MLP}_*(\mathbf{C}_{\text{point}})\right), \quad * \in \{\text{scale, shift, select}\} \\
\widehat{\mathbf{X}}^{n-1} &= \mathbf{I}_{\text{select}} \odot \text{SelfAttention}\left(\mathbf{I}_{\text{scale}} \odot \text{LayerNorm}(\mathbf{X}^{n-1}) + \mathbf{I}_{\text{shift}}\right) + \mathbf{X}^{n-1}, \\
\widehat{\mathbf{I}}_* &= \text{Concat}\left(\widehat{\text{MLP}}_*(\mathbf{C}_{\text{domain}}).\texttt{repeat}, \widehat{\text{MLP}}_*(\mathbf{C}_{\text{point}})\right), \quad * \in \{\text{scale, shift, select}\} \\
\mathbf{X}^n &= \widehat{\mathbf{I}}_{\text{select}} \odot \text{FeedForward}\left(\widehat{\mathbf{I}}_{\text{scale}} \odot \text{LayerNorm}(\widehat{\mathbf{X}}^{n-1}) + \widehat{\mathbf{I}}_{\text{shift}}\right) + \widehat{\mathbf{X}}^{n-1},
\end{aligned}
\tag{2}
$$

where $n \in \{1, \ldots, N\}$, and $\mathbf{X}^n$ is the output of the $n$-th layer. Since the PDE components have a crucial impact on the range of the output, we *scale* and *shift* $\mathbf{X}^N$ based on the deep equation conditions, and then linearly project $\mathbf{X}^N$ to obtain the final output as predictions of $\mathbf{Y}$.

## 4   EXPERIMENTS

We conduct extensive experiments to evaluate Unisolver on three challenging large-scale benchmarks, covering a wide range of PDE components and diverse generalization scenarios.

**Benchmarks**   As summarized in Table 2, three experimental large-scale benchmarks cover varied dimensions, resolutions and PDE components. The HeterNS is an extension of the NS dataset from FNO (2021a), incorporating multiple viscosity coefficients and external forces to enhance diversity. The 1D time-dependent PDEs, introduced by PDEformer (2024), is a large-scale dataset containing three million structured 1D PDE samples and evaluate the zero-shot generalization performance on PDEBench (2022). The 2D mixed PDEs, collected by DPOT (2024), include 12 diverse datasets from four well-established benchmarks. More details can be found in Appendix F.

Table 2: Summary of benchmarks. #GPU hours are calculated by averaging the training time of all models on one A100 GPU. Detailed compute resources can be found in Appendix H.7. ✓ indicates the PDE component will change among different samples, while × refers to unchanged ones.

| Benchmarks | #Dim | #Resolution | #Samples | #GPU hours | Symbols | Coefficient | Force | Geometry | Boundary |
|---|---|---|---|---|---|---|---|---|---|
| HeterNS | 2D+Time | (64,64,10) | 15k | ∼60h | × | ✓ | ✓ | × | × |
| 1D time-dependent PDEs | 1D+Time | (256,100) | 3M | ∼3000h | ✓ | ✓ | ✓ | × | ✓ |
| 2D mixed PDEs | 2D+Time | (128,128,10) | 74.1k | ∼800h | ✓ | ✓ | ✓ | ✓ | ✓ |

**Baselines**   We compare Unisolver with six advanced baselines on the HeterNS to demonstrate its generalizability under varied PDE components: the well-established FNO (2021a), PINO (2021b) and ViT (2020) and current state-of-the-art methods Factformer (2023b), ICON (2023) and MPP (2023). We augment these baselines by providing sufficient physics information to ensure a fair comparison, either by concatenating the inputs with varied PDE components, providing prompting trajectories (ICON) or applying physics-informed loss (PINO). Furthermore, we compare Unisolver with two generalizable solvers—PDEformer (2024) and DPOT (2024) on zero-shot generalization performance in a head-to-head manner. We refrain from including additional baselines on these two benchmarks due to the substantial computational cost of using million-scale samples.

Table 3: Performance comparison (relative L2) on HeterNS with different viscosity coefficients and fixed force frequency coefficient $\omega = 2$. For clarity, the best result is in *bold* and the second best is *underlined*. Promtotion refers to the relative improvement over the second-best method.

| HeterNS | Viscosity / Params | In-distribution Test | | | | | Zero-shot Generalization | | | | | |
|---|---|---|---|---|---|---|---|---|---|---|---|---|
| | | $\nu$ = 1e-5 | $\nu$ = 5e-5 | $\nu$ = 1e-4 | $\nu$ = 5e-4 | $\nu$ = 1e-3 | $\nu$ = 8e-6 | $\nu$ = 3e-5 | $\nu$ = 8e-5 | $\nu$ = 3e-4 | $\nu$ = 8e-4 | $\nu$ = 2e-3 |
| FNO | 4.7M | 0.0669 | 0.0225 | 0.0114 | 0.0031 | 0.0011 | 0.0702 | 0.0373 | 0.0141 | 0.0088 | 0.0084 | 0.2057 |
| PINO | 4.7M | 0.1012 | 0.0443 | 0.0263 | 0.0073 | 0.0031 | 0.1014 | 0.0646 | 0.0299 | 0.0142 | **0.0081** | 0.1894 |
| ViT | 4.8M | 0.0432 | 0.0206 | 0.0098 | 0.0031 | 0.0015 | 0.0458 | 0.0353 | 0.0119 | 0.0100 | 0.0174 | 0.1878 |
| Factformer | 5.1M | 0.0571 | 0.0259 | 0.0148 | 0.0018 | 0.0010 | 0.0489 | 0.0642 | 0.0167 | 0.1808 | 0.0639 | 0.3224 |
| ICON | 4.5M | 0.0585 | 0.0267 | 0.0144 | 0.0054 | 0.0029 | 0.0606 | 0.0387 | 0.0169 | 0.0246 | 0.0110 | 0.2149 |
| MPP | 4.9M | 0.0775 | 0.0496 | 0.0321 | 0.0098 | 0.0043 | 0.0796 | 0.0648 | 0.0376 | 0.0387 | 0.0236 | 0.2595 |
| **Unisolver** | 4.1M | **0.0321** | **0.0094** | **0.0051** | **0.0015** | **0.0008** | **0.0336** | **0.0178** | **0.0064** | **0.0066** | 0.0096 | **0.1504** |
| Promotion | / | 25.7% | 54.4% | 48.0% | 16.7% | 20.0% | 26.6% | 49.6% | 46.2% | 25.0% | / | 19.9% |

**Implementations**   All methods in the HeterNS benchmark are trained for 300 epochs using relative L2 loss and the ADAM optimizer (Kingma & Ba, 2015) with an initial learning rate of 0.0005 and a cosine annealing learning rate scheduler (Loshchilov & Hutter, 2016). The batch size is set to 60. For the 1D time-dependent PDEs and 2D mixed PDEs, we follow the training strategies from PDEformer (2024) and DPOT (2024) to ensure a fair comparison. Relative L2 is used as the evaluation metric. See Appendix G for full implementation details and hyper-parameter configurations. The inference code and checkpoint for 1D PDEs have been provided in supplementary materials.

## 4.1 HETEROGENEOUS 2D NAVIER-STOKES EQUATION (HETERNS)

**Setups**   We introduce HeterNS, an extension of the widely used 2D NS dataset (Li et al., 2021a), to assess how models handle diverse PDE components, particularly viscosity coefficients and force terms. It comprises five viscosity coefficients $\nu$ and three force terms differentiated by frequency $\omega$, resulting in 15 combinations of PDE components and 15,000 training samples. As depicted in Figure 4, we evaluate the model performance on *in-distribution test* with only unseen initial conditions and *zero-shot generalization* involving both unseen initial conditions and variations in viscosity coefficients or force terms.

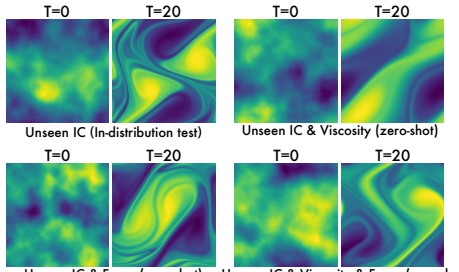

Figure 4: Visualization of various evaluation scenarios on the HeterNS benchmark.

**Results**   As shown in Tables 3-4, Unisolver achieves the best performance in 10 of 11 tasks, covering both in-distribution test and zero-shot generalization settings. It is worth noting that external force generalization is a highly difficult task, as the force term fundamentally determines the fluid evolution patterns. Still, Unisolver surpasses other methods in this challenging task, with significantly greater promotions in zero-shot generalization settings (**average 43.9%**) than in-distribution test settings (**average 27.4%**), demonstrating the effectiveness of our design in capturing generalizable physics relations between external force and model inputs. Even though we explicitly concatenate the varied PDE components with the model inputs, most advanced neural operators perform poorly on HeterNS. Specifically, all compared neural operators fail to solve the case of $\omega = 0.5$ in Table 4 with the relative error exceeding 0.5, further highlighting the generalizability of Unisolver. We also include experiments in Appendix H.1, where both viscosity and force are unseen. Unisolver still achieves considerable improvement (**average 41.3%**) on this challenging double unseen setting.

## 4.2 1D TIME-DEPENDENT PDES

**Setups**   This benchmark contains three million high-quality 1D time-dependent PDE samples with varying equation formulations, coefficients, force terms and boundary conditions. We perform joint training on this extensive dataset, where the input for the training task includes all relevant PDE components, and the output records full space-time fields. After training, the model is evaluated across multi-

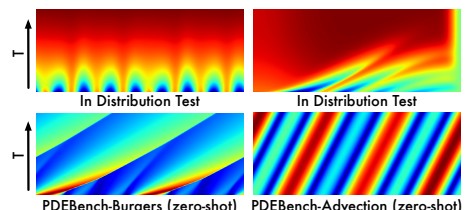

Figure 5: Showcases from various evaluation scenarios on the 1D time-dependent PDEs.

Table 4: Comparison (relative L2) on HeterNS with varied force and fixed viscosity $\nu = 10^{-5}$.

| HeterNS | Force Params | In-distribution Test | | | Zero-shot Generalization | | | |
|---|---|---|---|---|---|---|---|---|
| | | $\omega = 1$ | $\omega = 2$ | $\omega = 3$ | $\omega = 0.5$ | $\omega = 1.5$ | $\omega = 2.5$ | $\omega = 3.5$ |
| FNO | 4.7M | 0.0640 | 0.0661 | 0.1623 | 1.1100 | 0.1742 | 0.1449 | 0.2974 |
| PINO | 4.7M | 0.0914 | 0.1012 | 0.2707 | 1.0570 | 0.5010 | 0.4660 | 0.8380 |
| ViT | 4.8M | 0.0348 | 0.0432 | 0.1000 | 0.7900 | 0.1412 | 0.1240 | 0.2080 |
| Factformer | 5.1M | 0.0409 | 0.0570 | 0.0982 | 0.8591 | 0.1207 | 0.1243 | 0.2047 |
| ICON | 4.5M | 0.0435 | 0.0585 | 0.1345 | 1.1950 | 0.5295 | 0.5009 | 0.8231 |
| MPP | 4.9M | 0.0596 | 0.0775 | 0.1620 | 0.5532 | 0.2224 | 0.2180 | 0.3803 |
| **Unisolver** | 4.1M | **0.0244** | **0.0321** | **0.0720** | **0.0980** | **0.0770** | **0.0720** | **0.1740** |
| Promotion | / | 29.9% | 25.7% | 26.7% | 82.3% | 36.2% | 41.9% | 15.0% |

Table 5: Comparison (relative L2) of *in-distribution test* and *zero-shot generalization* on 1D time-dependent PDEs. Viscosity $\nu$ and advection velocity $\beta$ are dominated components of target PDEs.

| 1D Time-dep endent PDEs | Tasks Params | In-distribution Test | Zero-shot Burgers | | | Zero-shot Advection |
|---|---|---|---|---|---|---|
| | | | $\nu = 0.1$ | $\nu = 0.01$ | $\nu = 0.001$ | $\beta = 0.1$ |
| PDEformer | 22M | 0.0225 | 0.00744 | 0.0144 | 0.0393 | 0.0178 |
| Unisolver | 19M | **0.0108** | **0.00513** | **0.00995** | **0.0299** | **0.0138** |
| Promotion | / | 52.0% | 31.0% | 30.9% | 23.9% | 22.5% |

ple test settings, including in-distribution test, as well as zero-shot generalization on the Burgers and Advection equations from PDEBench (Takamoto et al., 2022), which is an another *unseen* dataset.

**Results** Table 5 presents that the in-distribution test performance of Unisolver is significantly better than that of PDEformer, indicating that our design of incorporating complete PDE components is more effective than the computational graph utilized by PDEformer in representing intricate physical relations. Additionally, Unisolver achieves better performance in four zero-shot generalization scenarios, with an average improvement of **27.1%** over PDEformer, even with fewer parameters.

### 4.3 2D MIXED PDEs

**Setups** This benchmark involves 12 datasets from four prominent benchmarks, covering a wide range of PDEs. After joint training on these diverse datasets, we perform in-distribution tests on each dataset. Notably, the in-distribution test set also involves challenging variations in the PDE components. Moreover, unlike the balanced data in HeterNS, these datasets exhibit significant imbalances across different PDE components. To mitigate this issue, we adopt the balanced data sampling method from DPOT (Hao et al., 2024); however, it still poses considerable challenges in managing such diverse PDE samples.

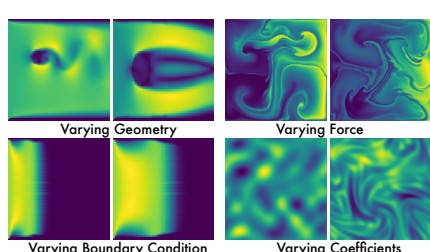

Figure 6: Showcases from in-distribution test sets on the 2D mixed PDEs.

**Results** As shown in Table 6, Unisolver outperforms DPOT (Hao et al., 2024) in 11 out of 12 in-distribution test sets with an remarkable average promotion of **17.5% (5.50→ 4.54)**, except for the small Diffusion-Reaction (DR) dataset whose relative L2 is less than 5%, verifying the effectiveness of our design in modeling such complex relations. Unisolver shows consistently superior performance in PDE component-dominated tasks, including coefficient generalization in FNO (2021a), force generalization in PDEArena (2023), and geometry generalization in CFDBench (2023), highlighting its ability to capture generalizable representations from complete PDE components.

### 4.4 MODEL ANALYSIS

**Ablations** As shown in Table 7, we further investigate the effect of LLM embeddings and condition modeling modules on 50,000 samples from the 1D time-dependent PDEs benchmark.

Firstly, in the LLM ablations, without LLM embedding, performance is the worst among all cases, even worse than replacing by orthogonal random vector. LLaMA-3 brings a **5.76%** averaged promotion compared to models without LLM embedding, indicating its essential role in learning PDEs.

Table 6: Performance comparison (relative L2 ($\times 10^{-2}$)) across 12 in-distribution test sets. For conciseness, we use a "source-PDE" format to denote different tasks (e.g. FNO-NS). The second row lists the primary PDE components considered for each dataset. See Appendix F.3 for details.

| 2D Mixed PDEs | Tasks | FNO-NS-$\nu$ | | | PDEBench-CNS-(M,$\zeta$) | | | | PDEBench | | PDEArena | | CFDBench-NS | Average Error |
|---|---|---|---|---|---|---|---|---|---|---|---|---|---|---|
| | Params | 1e-5 | 1e-4 | 1e-3 | (1, 0.1) | (1, 0.01) | (0.1, 0.1) | (0.1, 0.01) | DR | SWE | NS | NS-Force | Geometry | |
| DPOT | 30M | 5.53 | 4.42 | 1.31 | 1.53 | 3.37 | 1.19 | 1.87 | **3.79** | 0.66 | 9.91 | 31.6 | 0.70 | 5.50 |
| Unisolver | 33M | **4.17** | **3.36** | **0.61** | **1.23** | **2.89** | **1.01** | **1.59** | 4.39 | **0.45** | **6.87** | **27.4** | **0.54** | **4.54** |
| Promotion (%) | / | 24.6 | 24.0 | 53.4 | 19.6 | 14.2 | 15.1 | 15.0 | / | 31.8 | 30.7 | 13.3 | 22.9 | 17.5 |

Table 7: Ablation results on the LLM embeddings and the Condition Modeling. Variants of the former include without LLM embeddings (w/o LLM) and replacing by orthogonal random vectors (Random Vector), and variants of the latter include without subspace decoupling (w/o Subspace) and directly concatenating components (Concat). "Unchanged" means no changes to the default design.

| Relative L2 1D Time-dependent PDEs | LLM Embeddings | Condition Modeling | In-distribution Test | Zero-shot Generalization | | | |
|---|---|---|---|---|---|---|---|
| | | | | Burgers $\nu = 0.1$ | Burgers $\nu = 0.01$ | Burgers $\nu = 0.001$ | Advection $\beta = 0.1$ |
| Unisolver Ablations | w/o LLM | Unchanged | 0.0295 | 0.0189 | 0.0692 | 0.1432 | 0.0637 |
| | Random Vector | Unchanged | 0.0290 | 0.0185 | 0.0675 | 0.1471 | 0.0632 |
| | Unchanged | w/o Subspace | 0.0287 | 0.0187 | 0.0675 | 0.1478 | 0.0625 |
| | Unchanged | Concat | 0.0317 | 0.0236 | 0.0802 | 0.1586 | 0.0732 |
| *final | Unchanged | Unchanged | **0.0277** | **0.0176** | **0.0659** | **0.1350** | **0.0603** |

Notably, since the LLM only encodes one of six components, the equation symbols, a promotion of around 5% is a significant margin. Moreover, we compare the Unisolver's performance across different language models in Figure 7, including LLaMA-3, LLaMA-2 and T5. The results are comparable, indicating each model possesses sufficient ability to encode prior mathematical information.

Secondly, in condition modeling ablations, removing subspace decoupling introduces interference between different groups of PDE conditions, significantly impairing performance in *zero-shot generalization settings*, with an average drop of **5.45%**. Moreover, direct concatenation of PDE components severely hinders relation learning (**21.0%** average drop), indicating the benefits of our design.

**Visualization of the learned PDE embeddings**     As depicted in Figure 7(b-c), we apply the principal component analysis (PCA) (Jolliffe & Cadima, 2016) to intuitively visualize the LLM embeddings of equation symbols and deep PDE conditions learned by Unisolver for 1D time-dependent PDEs. In Figure 7(b), we observe that PDEs with similar complexity are encoded into similar embeddings, highlighting that LLM can indeed effectively capture prior mathematical information. In

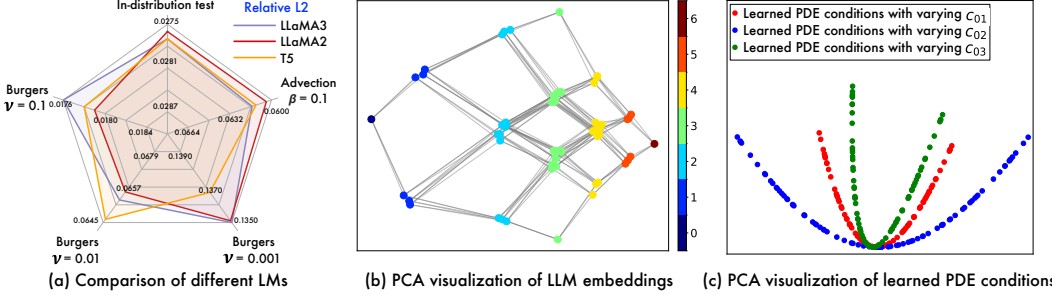

(a) Comparison of different LMs     (b) PCA visualization of LLM embeddings     (c) PCA visualization of learned PDE conditions

Figure 7: (a) Comparison of different language models. (b) PCA visualization of LLM embeddings. The considered PDE family contains six coefficients, such as $c_{01}, c_{02}, c_{03}$. Different colors represent the number of non-zero coefficients, intuitively indicating the complexity of PDEs. A zero coefficient results in the removal of a term from the equation, impacting the representations embedded by the LLM. (c) PCA visualization of learned deep PDE conditions, $\mathbf{I}_{select}$ in Eq. (2). We vary only one coefficient at a time and keep the others fixed at zero, forming the shown parabolic-like trajectories.

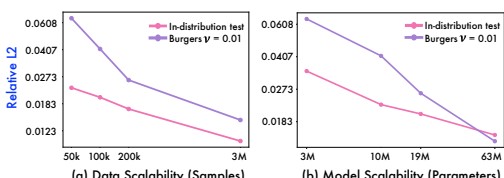

Figure 8: Capability of Unisolver to handle partial-observed data simulated on the HeterNS.

Figure 7(c), the trajectories of deep conditions resemble parabolas with varying degrees of curvature, indicating that the learned deep conditions successfully capture the variations of PDE components.

**Incomplete component scenario** In real-world applications, we may lack complete PDE components. To demonstrate the capability of Unisolver in handling such situations, we randomly replace PDE components with learnable tokens at a 30% probability in the HeterNS benchmark to simulate partially observed real-world data. For inference, we can flexibly choose whether to provide PDE components as inputs. As shown in Figure 8, even with incomplete components, Unisolver surpasses FNO (2021a) in most cases, especially in more complex cases with lower viscosity coefficients. Moreover, complete PDE information further improves the model's performance (**average 21.6%**), supporting our motivation that complete information is essential for PDE solving.

**Scalability** Scalability is crutial for building a universal neural PDE solver. Figure 9 illustrates Unisolver's scalability, where we progressively increase the *training data* by 60 times and the *model parameters* by 21 times. Unisolver exactly displays the scaling law, achieving better performance with increased data and parameters, posing the potential for a practically universal neural PDE solver.

Figure 9: Data scalability (60x) and model scalability (21x) on the 1D time-dependent PDEs. Relative L2 results are plotted on a log-log scale.

**Case study** To provide a clear comparison, we provide showcases on the HeterNS in Figure 10. All the presented trajectories are generated from the same initial condition but exhibit distinct final fields, underscoring the determining role of PDE components. Further, we observe that Unisolver significantly outperforms FNO under complex conditions, such as smaller viscosity $\nu$ and larger force coefficient $\omega$, particularly in zero-shot generalization settings. More showcases can be found in Appendix D.

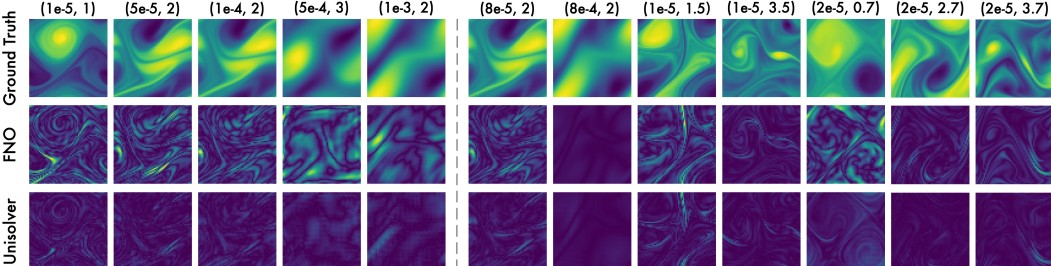

Figure 10: Error maps (the absolute difference between model predictions and ground truth) of FNO and Unisolver on the HeterNS, where all cases share the same initial condition but differ in *viscosity* ($\nu$) and *force* ($\omega$) (shown in the first row by the pairs ($\nu, \omega$)). The left panel shows in-distribution tests, while the right panel shows zero-shot generalization settings.

## 5 CONCLUSION

To break the generalization bottleneck, this paper presents Unisolver as a PDE-conditional Transformer, which stems from the theoretical analysis of the PDE-solving process. Concretely, Unisolver identifies and systematically encodes a complete set of PDE components into domain-wise and point-wise deep conditions separately and specifically. By integrating these conditions with Transformers through a decoupled mechanism, Unisolver can handle universal PDE components and achieve consistent state-of-the-art results across three challenging, large-scale benchmarks. Extensive analyses are provided to verify the effectiveness, generalizability and scalability of our model.

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

## A FINE-TUNING PERFORMANCE

Zero-shot generalization serves as a valuable metric, but in scenarios where datasets differ substantially from the training set, the model's zero-shot performance may be limited. In such instances, *fine-tuning* performance is critical, since it reflects the model's ability to learn fundamentally generalizable representations through large-scale training. We present Unisolver's fine-tuning performance on 1D time-dependent and 2D mixed PDEs in Figures 11-12, with 100 epochs for 1D time-dependent PDEs and 200 epochs for 2D mixed PDEs, both amounting to *20% of the total training epochs from scratch*, demonstrating fast adaptation.

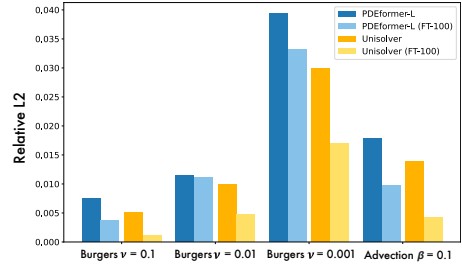

For 1D time-dependent PDEs, as shown in Figure 11, fine-tuning 100 epochs on the Burgers and Advection equations from PDEBench (Takamoto et al., 2022) significantly enhances Unisolver's performance, reducing error by **61%** compared to zero-shot results and achieving a **59.3%** improvement over PDEformer (Ye et al., 2024) under the same fine-tuning conditions. These results prove the condition modeling in Unisolver is more effective than the computational graph proposed by PDEformer, especially for fast adaptation. For 2D mixed PDEs, as shown in Figure 12, after 200 epochs of fine-tuning for each

Figure 11: Fine-tuning performance on 1D time-dependent PDEs. "FT-100" means fine-tuning on each dataset for 100 epochs.

dataset, Unisolver reduces error by more than **12%** compared to zero-shot generalization performance and outperforms DPOT (Hao et al., 2024) under the same fine-tuning conditions by **14%**, showcasing its ability to extract generalizable knowledge from diverse training datasets.

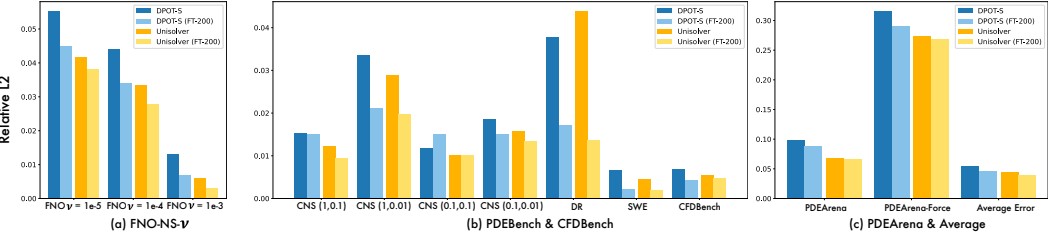

Figure 12: Performance comparison (relative L2) on 2D mixed PDEs after 200 epochs of fine-tuning.

## B MORE ABLATIONS ABOUT LLM EMBEDDINGS

To further verify the role of LLM embeddings in encoding PDE information, we conduct three more additional ablation experiments. In the first experiment, the LLM only encodes the number of non-zero terms in the 1D PDE. In the second experiment, the LLM encodes the "wrong" PDE information. Specifically, we replace "*" with "/" and adjust polynomial orders to their reciprocals. For example, the original latex code $u_t + c_{01} * u + c_{02} * u^2 + s(x) + (c_{11} * u + c_{13} * u^3)_x = 0$ is transformed into $u_t + c_{01}/u + c_{02}/u^{1/2} + s(x) + (c_{11}/u + c_{13}/u^{1/3})_x = 0$. In the third experiment, we manually construct a one-hot vector for each PDE term and combining them to represent a full PDE. Then the combined one-hot vector is directly used by Unisolver without being encoded by an LLM. The results of these three ablation studies are shown in Table **??**.

The results indicate that the model indeed obtains additional information beyond merely the count of non-zero terms from the LLM embeddings. Moreover, embedding "wrong" mathematical information generally leads to a decline in performance, highlighting the importance of accurately embedding the PDE information. While we cannot definitely claim that the LLM "understands" mathematical knowledge, we can confirm that the use of LLM enables us to encode useful mathematical information into deep representations. Besides, we observe that the LLM embedding case consistently outperforms the manually constructed representation case in both in-distribution tests and four zero-shot generalization settings, showing a **4.23%** average improvement. Although the manually constructed representation aims to preserve the mathematical structure of the PDE as much as possible, the handcrafted features struggle to perfectly capture the mathematical structure provided by LLMs visualized in Figure 7, leading to a decrease in performance.

Table 8: *More ablations about LLM embeddings.* We include three more ablations to further demonstrate the rationale for using LLM embeddings. Relative L2 loss is reported.

| Ablation type | In-distribution Test | Burgers $\nu = 0.1$ | Burgers $\nu = 0.01$ | Burgers $\nu = 0.001$ | Advection $\beta = 0.1$ |
|---|---|---|---|---|---|
| Number of non-zero terms encoded by LLM | 0.0285 | 0.0180 | 0.0665 | 0.1391 | 0.0618 |
| "Wrong" expression encoded by LLM | 0.0289 | 0.0181 | 0.0672 | 0.1361 | 0.0619 |
| Manually constructed representation | 0.0282 | 0.0184 | 0.0675 | 0.1386 | 0.0679 |
| **Ours** | **0.0277** | **0.0176** | **0.0659** | **0.1350** | **0.0603** |

## C   MORE EXPERIMENTS ABOUT GENERALIZABILITY

We conduct two additional experiments to evaluate the generalization capability of Unisolver: first, we verify the benefits of joint training on different types of PDEs rather than training on them independently; second, we evaluate Unisolver's capability to generalize to new types of PDEs.

### C.1   THE BENEFIT OF JOINT TRAINING

We design a new experiment to evaluate the benefit of joint training on the 1D time-dependent PDE benchmark. As stated in Appendix F, the general equation formulations used in this benchmark include two polynomials, $f_0$ and $f_1$, both with a maximum order of 3. We construct three distinct sub-datasets, each with 10,000 samples, to test the impact of joint training. The polynomials in each dataset are fixed to orders of 1, 2, and 3, respectively, ensuring that the PDEs contained in these three datasets do not overlap. For instance, in the dataset with polynomials of order 3, only $c_{03}$ and $c_{13}$ are non-zero terms, while $c_{01}$, $c_{02}$, $c_{11}$ and $c_{12}$ are fixed to zero. We conduct both joint training and independent training for 500 epochs on these 3 subdatasets. The results are shown in the Table 9.

Table 9: *The benefit of joint training.* We consider three distinct subset, where the polynomials are fixed to orders of 1, 2 and 3, respectively. The performance (relative L2) of the joint training model is compared against the same model trained on each subset independently.

| Polynomial order | 1 | 2 | 3 |
|---|---|---|---|
| Independent Training | 0.0792 | 0.1161 | 0.1236 |
| **Joint Training** | **0.0555** | **0.0738** | **0.0695** |
| Promotion | 29.9% | 36.5% | 43.7% |

### C.2   EQUATIONS GENERALIZATION VIA FINETUNING

We design a new equation generalization scenario based on the 1D time-dependent PDEs benchmark. As stated in Appendix F, the general equation formulations used in this benchmark include two polynomials, $f_0$ and $f_1$, both with a maximum order of 3. We pretrain Unisolver on 50,000 samples of PDEs with polynomial orders of up to 2, and then fine-tune it for 200 epochs on PDEs with polynomial orders of 3. The fine-tuned model is compared against the same model trained from scratch for 500 epochs, with relative L2 error reported in Table 10. Results indicate that Unisolver pretrained on equations of polynomial order up to 2 can be efficiently fine-tuned to handle equations of polynomial order 3. Unisolver demonstrates strong generalization capabilities to unseen PDEs, significantly reducing the need for large training datasets when addressing new equations.

Table 10: *Generalization to unseen equations.* Unisolver is initially trained on equations with a polynomial order of up to 2, and subsequently fine-tuned for 200 epochs on equations with a polynomial order of 3. The performance (relative L2) of the fine-tuned model is compared against the same model trained from scratch for 500 epochs.

| Finetuning Examples | 5000 | 10000 | 20000 |
|---|---|---|---|
| Unisolver-from-scratch-500 | 0.3308 | 0.1913 | 0.1327 |
| **Unisolver-fine-tune-200** | **0.1624** | **0.1036** | **0.0891** |
| Promotion | 50.9% | 45.8% | 32.9% |

# D    MORE SHOWCASES

We provide additional showcases here to supplement the numerical results presented in the main text. First, we visualize the in-distribution test and zero-shot generalization cases on the HeterNS dataset in Figure 13 and Figure 14, respectively. Next, we present visualizations for 1D time-dependent PDEs in Figure 15. Finally, we illustrate the 12 diverse datasets from 2D mixed PDEs in Figure 16.

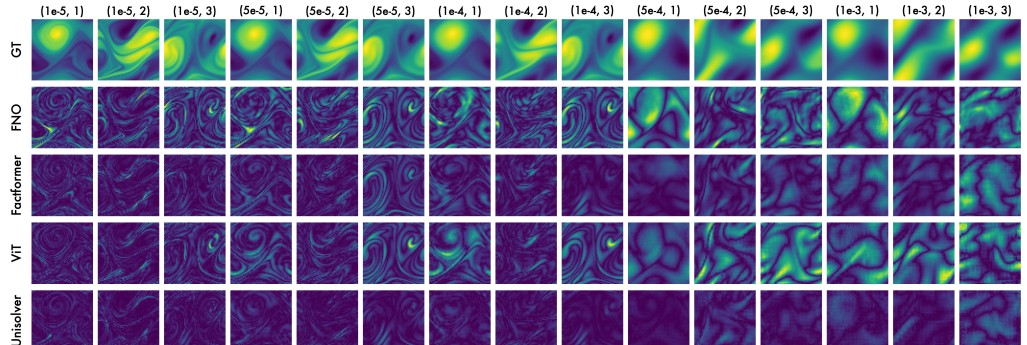

Figure 13: Error maps (the absolute difference between model predictions and ground truth) for in-distribution tests with *top three baselines* on the HeterNS dataset. See Table 3 and 4 for numerical comparison (relative L2). All data has the same initial condition and differs in *viscosity* ($\nu$) and *force* ($\omega$) (shown in the first row by the pairs ($\nu, \omega$)). Unisolver achieved the best visual performance among the compared baselines.

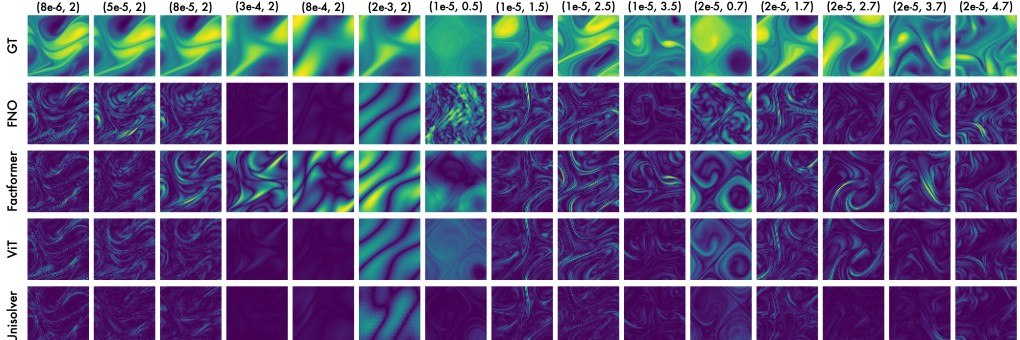

Figure 14: Error maps for zero-shot generalization settings with top three baselines on the HeterNS dataset with the same initial conditions and differs in *viscosity* ($\nu$) and *force* ($\omega$) (shown in the first row by the pairs ($\nu, \omega$)). See Table 3 and 4 for numerical comparison.

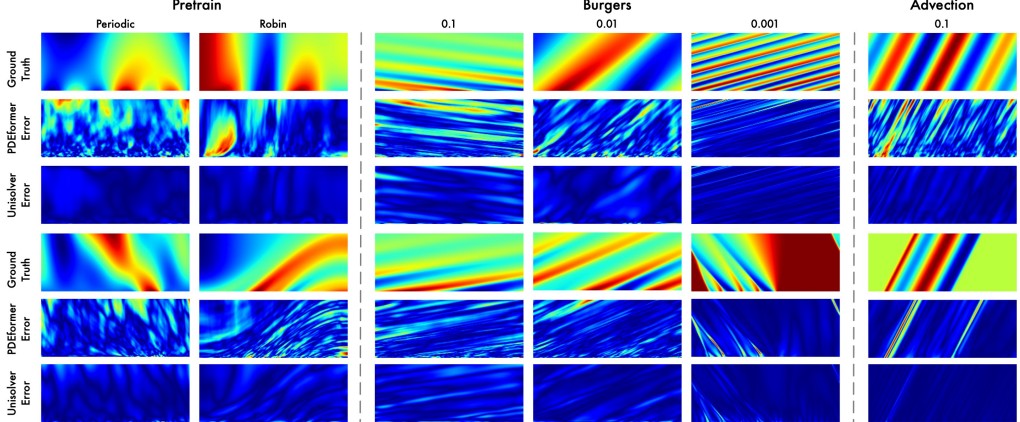

Figure 15: Error maps on the in-distribution test and zero-shot generalization (Burgers and Advection equation from PDEBench (Takamoto et al., 2022)) settings in 1D time-dependent PDEs. See Table 5 for numerical comparison. We visualize two cases: periodic boundary conditions and Robin boundary conditions in in-distribution tests. The number in the Burgers columns is the diffusion coefficient $\nu$ while the number in the Advection column is the advection speed $\beta$.

Figure 16: Unisolver predictions and error maps on 2D mixed PDEs. See Table 6 for numerical comparison with DPOT. Both predictions and error maps are provided. As shown in the CFDBench-NS columns, Unisolver presents an impressive ability to handle different geometry conditions.

# E  ANALYTICAL SOLUTION FOR THE STRING VIBRATION EQUATION

The solution of Eq. (1a) with boundary conditions (1b) and initial conditions (1c) is

$$u(x,t) = \frac{1}{2}\underbrace{(\Phi(x+at) + \Phi(x-at))}_{\text{Initial position}} + \frac{1}{2a}\underbrace{\int_{x-at}^{x+at}\Psi(\xi)\,d\xi}_{\text{Initial velocity}} + \frac{1}{2a}\underbrace{\int_0^t d\tau \int_{x-a(t-\tau)}^{x+a(t-\tau)}\underbrace{f(\xi,\tau)}_{\text{Force}}\,d\xi}_{\text{Geometry}},$$

(3)

where $\Phi(x)$, $\Psi(x)$ and $F(x,t)$ are odd, periodic functions with period $2L$ defined on the upper half plane, extended from $\phi(x)$, $\psi(x)$ and $f(x,t)$. The boundary conditions will be explicit by extending the equation to the upper half plane and solving it by operator splitting and characteristic lines.

Detailed proof can be found in (Evans, 2022) or other relevant books.

# F  BENCHMARKS

We provide a detailed description of the three large-scale benchmarks in our experiments here: a challenging, self-generated heterogeneous 2D Navier-Stokes Equations dataset (HeterNS) and two large-scale benchmarks, one proposed by PDEformer (Ye et al., 2024), and the other collected by DPOT (Hao et al., 2024). These benchmarks cover a wide range of PDEs and diverse generalization scenarios, which can test the generalizability of PDE solvers well.

## F.1  HETERNS

Similar to FNO (Li et al., 2021a), we consider the 2D Navier-Stokes equation in vorticity formulation for the viscous, incompressible fluid on a unit torus. We consider both in-distribution test and zero-shot generalization settings on HeterNS. See Figure 13 and 14 for a visual representation.

$$\partial_t w(x,t) + u(x,t)\cdot\nabla w(x,t) = \nu\Delta w(x,t) + f(x), \quad x\in(0,1)^2,\, t\in(0,T]. \quad (4a)$$
$$\nabla\cdot u(x,t) = 0, \qquad\qquad x\in(0,1)^2,\, t\in[0,T]. \quad (4b)$$
$$w(x,0) = w_0(x), \qquad\qquad x\in(0,1)^2. \quad (4c)$$

**Train set**  The problem involves two key PDE components: the viscosity coefficient and the force term. We experiment with viscosity coefficients $\nu \in [8\times 10^{-6}, 2\times 10^{-3}]$ and force terms in the form $f(x) = 0.1(\sin(\omega\pi(x_1 + x_2)) + \cos(\omega\pi(x_1 + x_2)))$. Specifically, our training set consists of $\nu \in \{1\times 10^{-5}, 5\times 10^{-5}, 1\times 10^{-4}, 5\times 10^{-4}, 1\times 10^{-3}\}$ and $\omega \in \{1, 2, 3\}$, resulting in 15 unique

combinations of PDE components. For each combination, we generate 1000 samples, yielding a total of 15,000 training samples. The dataset can be accessed at the following anonymous link.[2]

**In-distribution test set**  For testing, we first evaluate the in-distribution test sets, each containing 200 samples. In this setting, only the initial conditions differ from the training dataset, while all other PDE components remain the same.

**Zero-shot generalization set**  Zero-shot generalization settings present much greater challenges, as both the initial conditions and the viscosity coefficient or force terms may be entirely unseen during training. We assess the model's zero-shot performance on 200 samples, offering a more rigorous test of its ability to learn generalizable representations.

## F.2  1D TIME-DEPENDENT PDEs

This benchmark is proposed by PDEformer (Ye et al., 2024). It contains 3 million high-quality 1D time-dependent PDEs with various equation components for training and then evaluates the model performance using in distribution test sets and zero-shot generalization performance on Burgers and Advection equation from PDEBench (Takamoto et al., 2022), which is another distinct benchmark. See Figure 15 for a visual representation.

**Train set**  The training dataset is generated by the following PDE family:

$$\partial_t u + f_0(u) + s(x) + \partial_x(f_1(u) - \kappa(x)\partial_x u) = 0, \ (x,t) \in [-1,1] \times [0,1]. \tag{5a}$$

$$u(0,x) = g(x), \ x \in [-1,1]. \tag{5b}$$

where $f_i(u) = c_{i1}u + c_{i2}u^2 + c_{i3}u^3, i = 0, 1$. Each coefficient $c_{ik}$ is set to zero with a probability of 0.5, and otherwise uniformly sampled from the interval $[-3, 3]$. The variables $\kappa(x)$ and $s(x)$ can be zero, constant or physical fields, which are all randomly sampled from pre-defined distributions, as detailed in PDEformer's original paper (Ye et al., 2024). The initial condition $g(x)$ is randomly generated within the family of trigonometric functions, a super-position of sinusoidal waves as, $u_0(x) = \sum_{k_i=k_1,\ldots,k_N} A_i \sin(k_i x + \phi_i)$, where $k_i = 2\pi n_i/L_x$ are wave numbers and $n_i \in \mathbb{N}$ are selected randomly in $[1, n_{\max}]$, which is same as the zero-shot generalization tasks from PDEBench (Takamoto et al., 2022).

The dataset includes both periodic and non-periodic boundary conditions, with 1.5 million samples each. For the non-periodic cases, the boundary condition type at *each endpoint* are randomly selected from three pre-defined types: Dirichlet, Neumann, and Robin. The Dirichlet conditions specify the solution value at the boundary, while the Neumann conditions set the derivative value at the boundary, and the Robin conditions are a linear combination of the Dirichlet conditions and Neumann conditions. Therefore, Dirichlet and Neumann boundary conditions are regarded as corner cases of the Robin conditions.

We now provide a summary from the perspective of the complete PDE components. The *domain-wise components* of the training dataset include equation symbolic expression, i.e. Eq. (5), boundary condition types, and coefficients in two polynomials $f_i$ while *the point-wise components* include the physical fields $s(x)$ and $\kappa(x)$, which are considered as force terms and boundary value functions. The input observations are the initial conditions, discretized spatially at a resolution of 256. The output is the final solution $u(x,t)$, discretized spatially at 256 and temporally at 100.

**Symbolic variations**  Additionally, there is one important aspect to consider regarding the symbolic variations of equation symbols. A zero coefficient in the two polynomials $f_i$ results in the removal of a term from the equation. If the physical fields $\kappa(x)$ or $s(x)$ are zero, the corresponding term is removed from the prompt. When $\kappa(x)$ is constant, it is replaced by $\kappa$ to more accurately reflect the constant value, and the same applies to $s(x)$. These symbolic variations directly affect the equation formulations further embedded by the LLM, resulting in $2^6 \times 3 \times 3 = 576$ types of LLM embeddings, corresponding to 576 distinct equation types.

---

[2]https://drive.google.com/drive/folders/142c518gF9DWDD9FOx7TvtEwaNb5nHZUa

**In-distribution test set**  We generate 10,000 samples strictly following the configurations of the training dataset to ensure that all PDE components are within the same distribution. However, being in the same distribution does not mean that they have been seen before. Given to the multitudinous PDE family, all PDE components, besides the equation symbols, can still exhibit significant variations, making in-distribution tests is also a highly challenging task.

**Zero-shot generalization set**  We employ the following two 1D PDE datasets from PDEBench (Takamoto et al., 2022) as zero-shot generalization tasks. All zero-shot generalization tasks follow periodic boundary conditions and the same initial condition family as the training dataset. The resolution of these samples is $1024 \times 201$. For each dataset, we use 1000 test samples. We downsample the spatial resolution of these datasets to 256 and maintain the temporal resolution unchanged. The zero-shot PDEs consist of the Burgers equation and the Advection equation.

**(1) Burgers equation**  Burgers equation, as the fundamental equation in fluid mechanics, models the non-linear behavior and diffusion process of fluid dynamics as:

$$\partial_t u(t,x) + \partial_x(u(t,x)^2/2) = \nu/\pi \partial_{xx} u(t,x), \; x \in (0,1), t \in (0,2]. \tag{6a}$$

$$u(0,x) = u_0(x), \; x \in (0,1). \tag{6b}$$

where $\nu$ is the diffusion coefficient. In our zero-shot generalization settings, the Burgers equation dataset consists of three subsets, distinguished by the diffusion coefficient: $\nu = 0.1, 0.01, 0.001$. The diffusion coefficient represents the intensity of fluid variation, with smaller values corresponding to more complex fluid dynamics.

**(2) Advection equation**  The advection equation models pure advection behavior without non-linearity, which can be formalized as:

$$\partial_t u(t,x) + \beta \partial_x u(x,t) = 0, \; x \in (0,1), t \in (0,2]. \tag{7a}$$

$$u(0,x) = u_0(x), \; x \in (0,1), \tag{7b}$$

where the constant advection speed $\beta$ and equation symbols are considered domain-wise components in this dataset. In our zero-shot generalization settings, we use an advection speed of $\beta = 0.1$. It is worth noting that the advection equation has an analytic solution, given by $u(t,x) = u_0(x - \beta t)$.

**Fine-tuning**  We also provide fine-tuning results on 1D time-dependent PDEs in Appendix A. Compared to zero-shot generalization, we fine-tune the model using an additional 9,000 samples while testing on the same 1,000 samples.

**Domain alignment**  Notably, the spatiotemporal domain of the equations in PDEBench is $[0,1] \times [0,2]$, whereas the training dataset uses the domain $[-1,1] \times [0,1]$. To directly infer from the model trained on 1D time-dependent PDEs, we need to align the spatiotemporal domains through *spatial-temporal coordinate transformations*, which will result in corresponding changes to the PDE components. Technically, the zero-shot PDEs after the coordinate transformation are given by:

- Burgers equation: $\partial_{t'} u + \partial_{x'}(2u^2) - \frac{8\nu}{\pi} \partial_{x'x'} u = 0$, where $t' = \frac{t}{2}, x' = 2x - 1$.

- Advection equation: $\partial_{t'} u + \partial_{x'}(4\beta u) = 0$, where $t' = \frac{t}{2}, x' = 2x - 1$.

### F.3 2D MIXED PDEs

This benchmark is collected by DPOT (Hao et al., 2024), which consists of the following 12 diverse subsets from 4 benchmarks. We only conduct *in-distribution tests* in the 2D mixed PDEs. Notably, the in- distribution test set also involves challenging variations in the PDE components. See Figure 16 for a visual representations.

**FNO-$\nu$ (Li et al., 2021a)**  This well-established benchmark considers the 2D Navier-Stokes equation for a viscous, incompressible fluid in vorticity form on the unit torus. The task is to estimate the vorticity field of the future ten timesteps on a regular $64 \times 64$ grid based on the initial ten timesteps observations of the vorticity field. The only varying PDE component in this dataset is the *viscosity coefficient*, which takes values from the set $\{1 \times 10^{-5}, 1 \times 10^{-4}, 1 \times 10^{-3}\}$. We use 1,000 instances

for the viscosity value $1 \times 10^{-5}$, 9,800 instances for $1 \times 10^{-4}$, and 1,000 instances for $1 \times 10^{-3}$ to pre-train or fine-tune our model. The remaining 200 instances are used for testing its performance. In in-distribution tests, the initial conditions vary across samples.

**PDEBench (Takamoto et al., 2022)** The following three subsets are derived from PDEBench (Takamoto et al., 2022), encompassing three distinct equations: the compressible Navier-Stokes equation (*CNS*), the diffusion-reaction equation (*DR*), and the shallow-water equation (*SWE*). All datasets considered in PDEBench adhere to periodic boundary conditions. The spatial resolution of this benchmark is $128 \times 128$.

**(1) The compressible Navier-Stokes equation** models compressible fluid dynamics, including phenomena such as shock wave formation and propagation. In this dataset, two dominant domain-wise components are considered: the *Mach number* ($M$) and *shear viscosity* ($\zeta$). The dataset includes four combinations of these components, represented as coefficient pairs ($M, \zeta$): $(1, 0.1), (1, 0.01), (0.1, 0.1), (0.1, 0.01)$. Each combination provides 9,000 instances for training and 200 for testing. The task involves predicting the next 11 timesteps of multiple physical fields—vorticity, pressure, and density—given the initial 10 timesteps of observations. In in-distribution tests, the initial conditions vary across samples.

**(2) The shallow-water equation**, derived from the general Navier-Stokes equations, models free-surface flow problems like coastal tides, storm surges, and shallow lake flows. This equation is formalized as,

$$\partial_t h + \nabla \cdot (h\boldsymbol{u}) = 0, \tag{8a}$$

$$\partial_t (h\boldsymbol{u}) + \nabla \cdot \left( \frac{1}{2} h\boldsymbol{u}^2 + \frac{1}{2} g_r h^2 \right) = -g_r h \nabla b. \tag{8b}$$

where $h$ describes the water depth, $b$ describes a spatially varying bathymetry, $g_r$ describes the gravitational acceleration, and $\nabla \cdot (h\boldsymbol{u})$ can be interpreted as the directional momentum. A key characteristic of this dataset is its *long prediction horizon*. The task of interest is to predict the future 91 timesteps of water depth based on the first 10 timesteps of observations. In in-distribution tests, the initial conditions vary across samples.

**(3) The 2D Diffusion-Reaction Equation** involves two non-linearly coupled variables, namely the activator $u = u(t, x, y)$ and the inhibitor $v = v(t, x, y)$. It is primarily applicable for modeling biological pattern formation, such as the development of animal coat patterns, skin pigmentation and cellular organization. This equation is formalized as,

$$\partial_t u = D_u \partial_{xx} u + D_u \partial_{yy} u + R_u. \tag{9a}$$

$$\partial_t v = D_v \partial_{xx} v + D_v \partial_{yy} v + R_v. \tag{9b}$$

where $D_u = 1 \times 10^{-3}$ and $D_v = 5 \times 10^{-3}$ are the diffusion coefficient for the activator and inhibitor, respectively, and $R_u = R_u(u, v)$ and $R_v = R_v(u, v)$ are the corresponding reaction functions for the activator and inhibitor, which are defined by the Fitzhugh-Nagumo equation as,

$$R_u(u, v) = u - u^3 - k - v, \tag{10a}$$

$$R_v(u, v) = u - v, \tag{10b}$$

where $k = 5 \times 10^{-3}$. The initial condition is generated as standard normal random noise $u(0, x, y) \sim \mathcal{N}(0, 1.0)$ for $x \in (-1, 1)$ and $y \in (-1, 1)$. The dataset is temporally discretized into $N_t = 101$. A key characteristic of this dataset is its *long prediction horizon*. The task of interest is to predict the future 91 timesteps of $u$ and $v$ given the initial 10 timesteps of observations. In in-distribution tests, the initial conditions vary across samples.

**PDEArena (Gupta & Brandstetter, 2023)** This well-established benchmark considers the velocity function formulation of the incompressible Navier-Stokes equations, which is widely used in real-world applications, such as fluid flow in pipes, aerodynamic simulations, and weather prediction models. This equation is formalized as,

$$\partial_t \boldsymbol{v} = -\boldsymbol{v} \cdot \nabla \boldsymbol{v} + \mu \nabla^2 \boldsymbol{v} - \nabla p + \boldsymbol{f}, \tag{11a}$$

$$\nabla \cdot \boldsymbol{v} = 0. \tag{11b}$$

where $v \cdot \nabla v$ represents convection, meaning the rate of change of $v$ along its own direction, $\mu \nabla^2 v$ is the viscosity, i.e.the diffusion or net movement of $v$, $\nabla p$ corresponds to the internal pressure, and $f$ represents the external buoyancy force. The inclusion of the incompressibility constraint $\nabla \cdot u = 0$ ensures mass conservation within the equations.

The spatial resolution of PDEArena is $128 \times 128$. This benchmark includes two subsets: one with a *fixed external force* and another with a *varied external force*. In the fixed-force subset, the initial conditions vary across samples and the task is to predict the next 4 timesteps of velocity based on the initial 10 timesteps of observations, with 3,100 samples used for training and 200 samples for testing. In contrast, the more complex varied-force subset, where the initial conditions and force terms vary across samples, requires predicting 46 future timesteps, with 6,500 samples for training and 650 samples for testing.

**CFDBench (Luo et al., 2023)**  We consider three important and representative fluid dynamics problems that provide a comprehensive evaluation of a method's ability to generalize to unseen PDE components. These problems are: (1) flow in a lid-driven cavity, (2) flow through a circular tube, and (3) flow around a cylinder. The equation is formalized as follows:

$$\partial_t(\rho\boldsymbol{u}) + \nabla \cdot (\rho\boldsymbol{u}^2) = -\nabla p + \nabla \cdot \mu(\nabla\boldsymbol{u} + \nabla\boldsymbol{u}^T), \tag{12a}$$
$$\nabla \cdot (\rho\boldsymbol{u}) = 0. \tag{12b}$$

where $\rho$ is the constant density, $\mu$ is the dynamic viscosity, $\boldsymbol{u} = (u, v)^T$ is the velocity field, and $p$ is the pressure.

In in-distribution test settings, flows are generated for each problem with different PDE components, which are a combination of three types: (1) *boundary conditions*, (2) fluid physical *coefficients* such as density and viscosity, and (3) the *geometry* of the field. The boundary conditions refer to the inlet velocity or movement velocity, depending on the specific case. Each type of PDE component corresponds to a distinct subset. In each subset, the corresponding PDE components are varied while other parameters remain constant. We mix the three subsets following DPOT's configuration (Hao et al., 2024), resulting in 9,000 training samples and 1,000 testing samples. The initial resolution is $64 \times 64$, which is then interpolated to $128 \times 128$. The task is to predict the next 10 timesteps of velocity given the first 10 timesteps of observations.

**Fine-tuning**  We also provide fine-tuning results on 2D mixed PDEs in Appendix A. Given the significant diversity across the 12 subsets, we fine-tune the model using a specific training subset to allow it to focus on the target subset and achieve improved performance.

# G  IMPLEMENTATION DETAILS

In this section, we provide a detailed description of the implementation, covering three key aspects: metrics, implementations for each benchmark and LLM embeddding details.

## G.1  LOSS AND METRICS

**Relative L2 for physics fields**  We can calculate the relative L2 distance between ground truth $u$ and model prediction $\hat{u}$ as follows:

$$\text{Relative L2 of } (u, \hat{u}) = \frac{\|u - \hat{u}\|_{L^2}}{\|u\|_{L^2}}. \tag{13}$$

where $\|u - \hat{u}\|_{L^2}$ is the $L^2$-distance between the predicted solution $\hat{u}$ and the ground-truth solution $u$, and $\|u\|_{L^2}$ is the $L^2$-norm of the ground-truth solution. Relative L2 is used as both training loss and evaluation metric.

**Relative Promotion**  Given the error of our model $\epsilon_{\text{ours}}$ and the error of the second best model $\epsilon_{\text{second-best model}}$, we can calculate the relative promotion as follows:

$$\text{Relative Promotion} = 1 - \frac{\epsilon_{\text{ours}}}{\epsilon_{\text{second-best model}}}. \tag{14}$$

Relative promotion is widely used in the comparison and analytical experiments across the three large-scale benchmarks to measure the improvement of the Unisolver relative to the base models.

**Relative Drop**  Given the error of our model $\epsilon_{\text{ours}}$ and the error of the ablation model $\epsilon_{\text{ablation model}}$, we can calculate the relative drop to quantify the extent of performance degradation in the ablation experiments as follows:

$$\text{Relative Drop} = \frac{\epsilon_{\text{ablation}}}{\epsilon_{\text{ours}}} - 1. \tag{15}$$

Relative drop is only used in the ablation experiments in Section 4.4 to quantify the performance loss caused by removing or replacing a specific module.

### G.2 IMPLEMENTATIONS FOR EACH BENCHMARK

**HeterNS**  As outlined in Section 4, all the baseline models are trained under the same training strategy. We train the model using one-step predictions and test the model in an autoregressive manner. Specifically, all the models are trained for 300 epochs using the relative L2 loss and the ADAM optimizer (Kingma & Ba, 2015) with an initial learning rate of 0.0005, along with a cosine annealing learning rate scheduler (Loshchilov & Hutter, 2016). The batch size is set to 60. After the training process, we use the checkpoint *from the last epoch* to evaluate the model performance.

We also provide the detailed model architecture hyperparameters in Table 11. We configure each model to align their model parameter numbers to ensure a fair comparison. Note that for MPP (2023), We utilize the parameter configuration of the tiny version containing approximately five million trainable parameters, which is comparable to Unisolver and other baselines.

The varying PDE components in this benchmark include the viscosity coefficient and the external force. This physics information is provided to each baseline in an explicit or implicit way to ensure a fair comparison. For FNO (2021a), ViT (2020), Factformer (2023b), and MPP (2023), we explicitly concatenate the viscosity coefficient and the external force to the model input along the channel dimension to ensure a fair comparison. As the viscosity coefficient is essentially a scalar, we repeat it along the spatial dimensions and then perform the channel-concatenating process. ICON (2023) is a special baseline which takes prompting trajectories as additional inputs to implicitly extract the physics information. Consequently, instead of providing the PDE components, we augment the input to ICON with five additional prompting trajectories with the same viscosity and external force as the target trajectory. Note that ICON also needs additional prompting trajectories when conducting evaluation. For PINO (2021b), we follow the experiment setting in the original paper and train the model with physics-informed loss as a soft regularization. The proportion of physics-informed loss with regard to data loss is set to 0.1.

**1D Time-dependent PDEs**  We compare Unisolver with PDEformer-L in the 1D time-dependent PDEs benchmark, evaluating their in-distribution test and zero-shot generalization performance. We also report the model performance after fine-tuning in Appendix A. The pre-training and fine-tuing configurations for Unisolver and the fine-tuing configurations for PDEformer are listed in Table 12.

Following PDEformer's training strategies, we train the model to predict the solution at specific spatial-temporal coordinates through an INR. After the pre-training process, we use the checkpoint *from the last epoch* to evaluate the model performance for the in-distribution test and zero-shot generalization test in Section 4.2. For fine-tuning tasks, we utilize the fine-tuning script provided in the original repository of PDEformer and set the finetuning epochs to 100 for a fair comparison.

The model we use to compare with PDEformer-L with contains 19M trainable parameters, which is comparable to the 22M parameters of PDEformer-L. The model scalability experiments in Section 4.4 also show model configurations with different number of trainable parameters. We progressively increase the Unisolver parameter from 3M to 63M, thus resulting in 4 different model configurations. We present the detailed configurations of these models in Table 13. Note that in this benchmark, we utilize an adapted version PolyINR (Singh et al., 2023) to decode the encoder output from the Transformer backbone.

Table 11: Model hyperparameters of Unisolver and all baselines on the HeterNS benchmark.

| Hyperparameter | Value | Description |
|---|---|---|
| **FNO** | | |
| modes | 12 | The truncation number of Fourier modes |
| channels | 64 | The number of channels in the hidden layers |
| depth | 4 | The number of Fourier Layers in the neural network |
| **PINO** | | |
| modes | 12 | The truncation number of Fourier modes |
| channels | 64 | The number of channels in the hidden layers |
| depth | 4 | The number of Fourier Layers in the neural network |
| **ViT** | | |
| Attention dim | 256 | The hidden dimension of the transformer attention layer |
| MLP dim | 256 | The hidden dimension of the transformer FFN layer |
| patch_size | 4 | The height and width of the ViT patches |
| n_head | 8 | The number of attention heads |
| dim_head | 32 | The hidden dimension of each attention heads |
| depth | 12 | The number of Transformer Blocks in the neural network |
| **Factformer** | | |
| dim | 128 | hidden dimension of the transformer |
| n_head | 12 | The number of attention heads |
| dim_head | 64 | hidden dimension of each attention heads |
| depth | 8 | The number of Transformer Blocks in the neural network |
| **ICON** | | |
| Attention dim | 256 | The hidden dimension of the transformer attention layer |
| MLP dim | 256 | The hidden dimension of the transformer FFN layer |
| patch_size | 4 | The height and width of the ViT patches |
| n_head | 8 | The number of attention heads |
| dim_head | 32 | The hidden dimension of each attention heads |
| depth | 12 | The number of Transformer Blocks in the neural network |
| prompting numbers | 5 | number of prompting trajectories |
| **MPP** | | |
| Embed dim | 192 | Dimension of internal representation |
| n_head | 3 | The number of attention heads |
| depth | 8 | The number of Transformer Blocks in the neural network |
| patch_size | 8 | The height and width of the ViT patches |
| **Unisolver** | | |
| Attention dim | 256 | The hidden dimension of the transformer attention layer |
| MLP dim | 256 | The hidden dimension of the transformer FFN layer |
| patch_size | 4 | The height and width of the Unisolver patches |
| n_head | 8 | The number of attention heads |
| dim_head | 32 | The hidden dimension of each attention heads |
| depth | 8 | The number of Transformer Blocks in the neural network |

**2D Mixed PDEs** We compare Unisolver with DPOT-S with comparable model parameters in the 2D mixed PDEs benchmark. The training hyperparameter and model configurations are presented in Table 14. Similar to the HeterNS benchmark, We train the model using one-step predictions and test the model in an autoregressive manner.

This benchmark includes multiple diverse PDEs, each including its unique PDE components as illustrated in Appendix F. For example, the viscosity coefficient is the varying PDE components in the FNO-$\nu$ benchmark, while the shallow-water equation does not include this PDE component. Therefore, we must notice Unisolver whether a PDE component exists in a certain benchmark. To do so, Specifically, we introduce a binary masking channel to represent the existence of a certain PDE component. For example, when a PDE component exists in a benchmark, we concatenate an "1" with this component, indicating that this component is a valid one. When this PDE component

Table 12: Pre-training and finetuning configurations on the 1D time-dependent PDE benchmark.

| Parameter | Value | Description |
|---|---|---|
| **Unisolver Training** | | |
| batch_size | 1024 | Total batchsize used in one iteration |
| learning_rate | 6e-4 | The initial learning rate for the optimizer |
| epochs | 500 | The total number of training epochs |
| loss_type | Relative-l2 | Use relative L2-Norm for pretraining |
| optimizer | Adam | The optimization algorithm |
| lr_scheduler | Cosine Annealing | The learning rate scheduler |
| **Unisolver Finetuning** | | |
| batch_size | 256 | Total batchsize used in one iteration |
| learning_rate | 1e-5 | The initial learning rate for the optimizer |
| epochs | 100 | The total number of training epochs |
| loss_type | Relative-l2 | Use relative L2-Norm for finetuning |
| optimizer | Adam | The optimization algorithm |
| lr_scheduler | Cosine Annealing | The learning rate scheduler |
| **PDEformer Finetuning** | | |
| batch_size | 80 | Total batchsize used in one iteration |
| learning_rate | 5e-6 | The initial learning rate for the optimizer |
| epochs | 100 | The total number of training epochs |
| loss_type | Relative-l2 | Use relative L2-Norm for finetuning |
| optimizer | Adam | The optimization algorithm |
| lr_scheduler | Cosine Annealing | The learning rate scheduler |
| warmup_epochs | 10 | Epochs to linearly increase the learning rate |

Table 13: Model configurations of Unisolver with different sizes.

| Parameter Count | Attention dim | MLP dim | Layers (Backbone) | Heads | Layers (INR) |
|---|---|---|---|---|---|
| 3M | 256 | 256 | 6 | 4 | 4 |
| 10M | 384 | 384 | 8 | 8 | 8 |
| 19M | 512 | 512 | 8 | 8 | 8 |
| 63M | 768 | 768 | 12 | 12 | 12 |

does not exist in a benchmark, we concatenate an "0" with it, indicating that this is an invalid one. While the LLM embedding can provide some indication of this information, it does not serve as the input to the encoders of other components. This binary mask, however, aids the encoders' learning and further clarifies the information without introducing significant computational overhead.

### G.3 DETAILS OF THE LLM EMBEDDINGS

Here we give a detailed description of the prompts we use to encode the equation symbols. We will also discuss the impact of expressing the same PDE using different notations or mathematically equivalent transformations.

Note that the pre-training dataset of PDEformer (Ye et al., 2024) contains the PDE family following the formulation:

$$\partial_t u + f_0(u) + s(x) + \partial_x(f_1(u) - \kappa(x)\partial_x u) = 0, \ (x,t) \in [-1,1] \times [0,1] \tag{16}$$

where $f_i(u) = c_{i1}u + c_{i2}u^2 + c_{i3}u^3, i = 0, 1$. Each $c_{ij}$ can be zero or non-zero. The source term $s(x)$ and the viscosity term $\kappa(x)$ can be zero, a non-zero constant or a non-uniform function. As stated in Section 3.2, we use the LaTeX code of the equation as a prompt, and the output from the last Transformer block of the LLM serves as the symbol embedding of the equation. Table 15 gives some concrete samples of the LaTex code we use. There are 576 different equation symbols in total in the PDEformer benchmark.

Table 14: Training configurations on the 2D mixed PDE benchmark.

| Parameter | Value | Description |
|---|---|---|
| **Unisolver Training Configurations** | | |
| batch_size | 320 | Total batchsize used in one iteration |
| learning_rate | 1e-3 | The initial learning rate for the optimizer |
| epochs | 1000 | The total number of training epochs |
| loss_type | Relative-l2 | Use relative L2-Norm for pretraining |
| optimizer | AdamW | The optimization algorithm |
| lr_scheduler | OneCycle | The learning rate scheduler |
| warmup_epochs | 200 | Epochs to linearly increase the learning rate |
| **Unisolver Model Configurations** | | |
| Attention dim | 768 | The hidden dimension of the transformer attention layer |
| MLP dim | 768 | The hidden dimension of the transformer FFN layer |
| patch_size | 8 | The height and width of the ViT patches |
| n_head | 8 | The number of attention heads |
| dim_head | 96 | The hidden dimension of each attention heads |
| depth | 6 | The number of Transformer Blocks in the neural network |

Table 15: Sample LaTeX codes for different equations used in the PDEformer benchmark.

| LaTeX Code of Differential Equations | Problem Description |
|---|---|
| `u_t + (c_{12} * u^2)_x = 0` | Inviscid Burgers Equation |
| `u_t + (c_{12} * u^2 + kappa * u_x)_x = 0` | Viscid Burgers Equation |
| `u_t + (c_{11} * u)_x = 0` | Advection Equation |
| `u_t + (c_{11} * u + kappa * u_x)_x = 0` | Advection-Diffusion Equation |
| `u_t + c_{01} * u + (kappa * u_x)_x = 0` | Reaction-Diffusion Equation |
| `u_t + c_{01} * u + c_{02} * u^2 + (c_{12} * u^2 + kappa * u_x)_x = 0` | Fisher-KPP Equation |
| `u_t + c_{01} * u + c_{02} * u^2 + c_{03} * u^3 + s(x) + (c_{11} * u + c_{12} * u^2 + c_{13} * u^3 + kappa(x) * u_x)_x = 0` | More General 1D Equations |

Note that a differential equation may have multiple equivalent representations, and different people may express the same equation differently. A potential solution is to design targeted prompts and employ advanced prompting techniques, such as chain of thought, to standardize these variations into a unified form, which is clearly within the capabilities of modern LLMs. This standardized form can then be used to enhance the learning of the solver.

# H  ADDITIONAL ANALYSES

## H.1  UNSEEN VISCOSITY AND UNSEEN EXTERNAL FORCE ON HETERNS

In addition to Tables 3 and 4, we further assess Unisolver's generalization on HeterNS compared to other baselines under more challenging conditions, where both the viscosity coefficient and external force are unseen. Specifically, we generate nine different component pairs $(\nu, \omega)$, each with 200 testing samples. Notably, one case features $\omega = 6$, which significantly exceeds the maximum value of $\omega = 3$ used during training, making it particularly difficult. The full results are presented in Table 16. Unisolver consistently outperforms all baselines, especially in the most challenging case with $\omega = 6$, with a relative promotion of 37.1%.

Table 16: Performance comparison (relative L2) on zero-shot generalization settings with unseen viscosity ($\nu$) and unseen force ($\omega$). The pairs in the first row are in the form of ($\nu, \omega$). For clarity, the best result is in bold and the second-best is underlined.

| L2RE | (2e-5, 0.7) | (2e-5, 1.7) | (2e-5, 2.7) | (2e-5, 3.7) | (2e-5, 4.7) | (4e-5, 0.8) | (4e-5, 1.4) | (4e-5, 2.3) | (4e-5, 6) |
|---|---|---|---|---|---|---|---|---|---|
| FNO | 0.1862 | 0.0640 | 0.1176 | 0.2404 | 0.4226 | 0.0873 | 0.1516 | 0.0655 | 1.3102 |
| PINO | 0.7002 | 0.2887 | 0.4776 | 0.8991 | 0.9187 | 0.3793 | 0.5596 | 0.3349 | 0.9634 |
| ViT | 0.1961 | 0.0690 | 0.1075 | 0.2057 | 0.2226 | 0.0488 | 0.1305 | 0.0772 | 0.2276 |
| Factformer | 0.2070 | 0.0720 | 0.0891 | 0.1594 | 0.1868 | 0.0892 | 0.1456 | 0.0618 | 0.2465 |
| ICON | 0.4729 | 0.3693 | 0.5202 | 0.8719 | 0.7891 | 0.2212 | 0.5112 | 0.3652 | 0.9058 |
| MPP | 0.4532 | 0.4029 | 0.5155 | 0.8421 | 0.8484 | 0.2961 | 0.4084 | 0.4801 | 1.0240 |
| Unisolver | **0.0781** | **0.0378** | **0.0471** | **0.1421** | **0.1364** | **0.0399** | **0.0433** | **0.0374** | **0.1431** |
| Promotion | 58.06% | 40.94% | 47.71% | 10.85% | 26.98% | 18.24% | 66.82% | 39.48% | 37.13% |

Table 17: Ablations with *different viscosity coefficient* $\nu$ and fixed force $\omega = 2$ on the HeterNS on removing some PDE components (*W/o*), and replacing domain-wise or point-wise conditions from our design to directly concat (*Concat*).

| HeterNS | Viscosity | In-distribution Test | | | | | Zero-shot Generalization | | | | |
|---|---|---|---|---|---|---|---|---|---|---|---|
| | Params | $\nu$ = 1e-5 | $\nu$ = 5e-5 | $\nu$ = 1e-4 | $\nu$ = 5e-4 | $\nu$ = 1e-3 | $\nu$ = 8e-6 | $\nu$ = 3e-5 | $\nu$ = 8e-5 | $\nu$ = 3e-4 | $\nu$ = 8e-4 |
| W/o viscosity | 4.1M | 0.0388 | 0.0127 | 0.0084 | 0.0031 | 0.0015 | 0.0410 | 0.0367 | 0.0099 | 0.0068 | 0.0119 |
| W/o force | 4.1M | 0.0353 | 0.0123 | 0.0074 | 0.0027 | 0.0017 | 0.0378 | 0.0198 | 0.0086 | 0.0096 | 0.0124 |
| Concat viscosity | 4.1M | 0.0343 | 0.0107 | 0.0058 | 0.0017 | 0.0011 | 0.0359 | 0.0192 | 0.0071 | 0.0278 | 0.0243 |
| Concat force | 4.1M | 0.0331 | 0.0103 | 0.0061 | 0.0018 | 0.0010 | 0.0357 | 0.0191 | 0.0071 | 0.0104 | 0.0101 |
| **Unisolver** | 4.1M | **0.0321** | **0.0094** | **0.0051** | **0.0015** | **0.0008** | **0.0336** | **0.0178** | **0.0064** | **0.0066** | **0.0096** |

## H.2 MORE ABLATION STUDIES ON PDE COMPONENTS AND CONDITIONAL MODELING

In addition to the ablation experiments presented in Table 7, we further conduct ablations on HeterNS to assess whether the proposed PDE information set is essential and whether the condition modeling is effective for the solver's learning. This is demonstrated by removing specific components and replacing Unisolver's condition modeling with direct concatenation of PDE information.

As shown in Tables 17 and 18, removing the information leads to a significant drop in performance compared to vanilla Unisolver, and concatenating the information directly also results in a huge decline. It is worth noting that the absence of external force information or its improper use (e.g. via direct concatenation) significantly degrades performance even in zero-shot viscosity generalization tasks, and vice versa, further highlighting the importance of including complete PDE components.

Table 18: Ablations with *different force* $\omega$ and fixed viscosity $\nu = 10^{-5}$ on the HeterNS on removing some PDE components (*W/o*), and replacing domain-wise or point-wise conditions from our design to directly concat (*Concat*).

| HeterNS | Force | In-distribution Test | | | Zero-shot Generalization | | | |
|---|---|---|---|---|---|---|---|---|
| | Params | $\omega$ = 1 | $\omega$ = 2 | $\omega$ = 3 | $\omega$ = 0.5 | $\omega$ = 1.5 | $\omega$ = 2.5 | $\omega$ = 3.5 |
| W/o viscosity | 4.1M | 0.0310 | 0.0388 | 0.0926 | 0.261 | 0.250 | 0.258 | 0.424 |
| W/o force | 4.1M | 0.0267 | 0.0353 | 0.0804 | 0.553 | 0.618 | 0.657 | 0.913 |
| Concat viscosity | 4.1M | 0.0265 | 0.0343 | 0.0786 | 0.1267 | 0.2057 | 0.2771 | 0.2689 |
| Concat force | 4.1M | 0.0259 | 0.0331 | 0.0764 | 0.5386 | 0.3392 | 0.2841 | 0.2753 |
| **Unisolver** | 4.1M | **0.0244** | **0.0321** | **0.0720** | **0.0980** | **0.0770** | **0.0720** | **0.1740** |

## H.3 LONG TRAJECTORY PREDICTION

We extend the temporal evolution steps of HeterNS to 30 steps, corresponding to 30 seconds of complex fluid dynamics, and report the zero-shot performance comparison between Unisolver and the

baselines in the Table 19. We present the performance on the subdataset with a viscosity coefficient of $\nu = 1 \times 10^{-5}$ and a force coefficient of $\omega = 2$. This is a particularly challenging task, as these models have never seen such long trajectories in the training data (at most 20 seconds). Despite this, Unisolver still achieves the best performance compared with the top three baselines.

Table 19: Zero-shot performance comparison (relative L2) with *top three baselines* about long trajectory prediction tasks (30 seconds) on the HeterNS.

|  | Unisolver | FNO | ViT | Factformer |
|---|---|---|---|---|
| Relative L2 | **0.1956** | 0.3105 | 0.2527 | 0.2962 |

## H.4 FULL SCALABILITY

As a supplement to Figure 9 in the main text, we also conduct experiments on different zero-shot generalization tasks from (Takamoto et al., 2022) and record the concrete data in Table 20 for clarity.

Table 20: Scalability results on in-distribution test sets and zero-shot generalization tasks, as depicted in Figure 9.

| L2RE | Data Scalability (Samples) | | | | Model Scalability (Parameters) | | | |
|---|---|---|---|---|---|---|---|---|
| Scale | 50k | 100k | 200k | 3M | 3M | 10M | 19M | 63M |
| In-distribution test | 0.0232 | 0.0202 | 0.0170 | 0.0106 | 0.0342 | 0.0226 | 0.0202 | 0.0156 |
| Zero-shot Burgers $\nu = 0.1$ | 0.0161 | 0.0116 | 0.0081 | 0.0051 | 0.0143 | 0.0134 | 0.0116 | 0.0091 |
| Zero-shot Burgers $\nu = 0.01$ | 0.0649 | 0.0412 | 0.0260 | 0.0144 | 0.0552 | 0.0421 | 0.0412 | 0.0351 |
| Zero-shot Burgers $\nu = 0.001$ | 0.1399 | 0.1003 | 0.0689 | 0.0299 | 0.1188 | 0.0976 | 0.1003 | 0.0889 |

## H.5 EFFICIENCY ANALYSIS

We provide the inference time and memory consumption for each model to predict a single frame on the HeterNS dataset, along with the calculation time and memory consumption for the numerical solver, which is used to generate the HeterNS dataset, to calculate the next frame, as summarized in the Table 21. The results are measured on an A100 GPU with a batch size of 1. Unisolver demonstrates comparable inference speed to FNO, while consuming less memory. Besides, all neural PDE solvers are approximately 1,000 times faster than the numerical solver, highlighting their potential as efficient surrogate models.

Table 21: *Efficiency Analysis.* The inference (calculation) time and memory consumption for each model and numerical solver to predict a single frame on the HeterNS dataset.

|  | FNO | PINO | ViT | Factformer | ICON | MPP | Unisolver | Numerical Solver |
|---|---|---|---|---|---|---|---|---|
| Average Inference (Calculation) Time / s | 0.0042 | 0.0042 | 0.0045 | 0.0103 | 0.0057 | 0.0120 | 0.0054 | 7.26 |
| Average Memory Usage / MB | 730 | 730 | 558 | 758 | 784 | 1200 | 554 | 524 |

## H.6 STANDARD DEVIATIONS

We repeat the experiments three times on the HeterNS benchmark and provide standard deviations here. As shown in Table 22-23, Unisolver surpasses the previous state-of-the-art models with high confidence. Note that we compare Unisolver with the *second-best model*, which is a strong baseline as it is not achieved by a single model. The results demonstrate that Unisolver significantly outperforms baseline models, with the second-best result falling more than three standard deviations behind, except in the case of viscosity $\nu = 8e - 4$.

Table 22: Standard Deviations on the HeterNS benchmark with different viscosity coefficients and fixed force frequency coefficient $\omega = 2$.

| Viscosity $\nu$ | In-distribution Test | | | | | Zero-shot Generalization | | | | |
|---|---|---|---|---|---|---|---|---|---|---|
| | $\nu = 1e\text{-}5$ | $\nu = 5e\text{-}5$ | $\nu = 1e\text{-}4$ | $\nu = 5e\text{-}4$ | $\nu = 1e\text{-}3$ | $\nu = 8e\text{-}6$ | $\nu = 3e\text{-}5$ | $\nu = 8e\text{-}5$ | $\nu = 3e\text{-}4$ | $\nu = 8e\text{-}4$ |
| Second-best model | 0.0432 | 0.0206 | 0.0098 | 0.0018 | 0.0010 | 0.0458 | 0.0353 | 0.0119 | 0.0088 | **0.0081** |
| **Unisolver** | **0.0321** | **0.0094** | **0.0051** | **0.0015** | **0.0008** | **0.0336** | **0.0178** | **0.0064** | **0.0066** | 0.0096 |
| Standard Deviation | ±0.0005 | ±0.0003 | ±0.0001 | ±0.0001 | ±0.00006 | ±0.0008 | ±0.0002 | ±0.0004 | ±0.0007 | ±0.00007 |
| Confidence Level | 99% | 99% | 99% | 99% | 99% | 99% | 99% | 99% | 99% | / |

Table 23: Standard Deviations on the HeterNS benchmark with different force ($\omega$) and fixed viscosity coefficient $\nu = 2$.

| Force $\omega$ | In-distribution Test | | | Zero-shot Generalization | | | |
|---|---|---|---|---|---|---|---|
| | $\omega = 1$ | $\omega = 2$ | $\omega = 3$ | $\omega = 0.5$ | $\omega = 1.5$ | $\omega = 2.5$ | $\omega = 3.5$ |
| Second-best Model | 0.0348 | 0.0432 | 0.0982 | 0.5532 | 0.1207 | 0.1240 | 0.2047 |
| **Unisolver** | **0.0244** | **0.0321** | **0.0720** | **0.0980** | **0.0770** | **0.0720** | **0.1740** |
| Standard Deviation | ±0.0003 | ±0.0002 | ±0.0003 | ±0.0015 | ±0.0048 | ±0.0051 | ±0.0021 |
| Confidence Level | 99% | 99% | 99% | 99% | 99% | 99% | 99% |

## H.7 DETAILED COMPUTE RESOURCES

Our models were trained on servers with *32 NVIDIA A100 GPUs*, each with 40GB memory. Here we present the compute resources in terms of GPU hours, where one GPU hour represents the time spent training on a single A100 GPU for one hour. This metric reflecting the resources required to reproduce the experimental results are shown in Table 24.

Table 24: Computational costs in GPU hours, measured on NVIDIA A100 GPUs (40 GB memory).

| Benchmarks | HeterNS | | | | | | | 1D Time-dependent PDEs | 2D Mixed PDEs |
|---|---|---|---|---|---|---|---|---|---|
| Models | FNO | Factformer | ViT | PINO | ICON | MPP | Unisolver | Unisolver | Unisolver |
| #GPU hours | 12 | 100 | 24 | 12 | 24 | 30 | 24 | 3000 | 800 |

## I FULL TRAJECTORY VISUALIZATIONS

To better understand the temporal evolution of the benchmark, we visualize the full trajectory of the ground truth and Unisolver predictions on HeterNS and 2D mixed PDEs. To further enhance clarity and provide a more intuitive understanding of these temporal dynamics, we have included videos to illustrate the trajectories frame by frame. Please refer to the supplementary material for details.

## J LIMITATIONS AND FUTURE WORK

This paper presents Unisolver to solve PDEs under universal PDE components, which achieves impressive performance supported by extensive analyses and visualizations. However, our method is currently limited to grid data due to the patchifying process during the embedding of point-wise components. Actually, this limitation is shared in all the generalizable PDE solvers, such as MPP (McCabe et al., 2023), Poseidon (Herde et al., 2024), PDEformer (Ye et al., 2024) and DPOT (Hao et al., 2024). One fundamental reason is the lack of suitable and large-scale irregular-mesh PDE datasets, which will require extremely high computation costs for generation and massive resources for collection. Since our primary focus in this paper is on the study of model architecture design and generalization capabilities, we would like to leave the irregular-mesh PDE dataset as a future work. Also, the capability to handle irregular meshes of Unisolver can be achieved by replacing the canonical Transformer with the latest geometry-general PDE solver: Transolver (Wu et al., 2024).

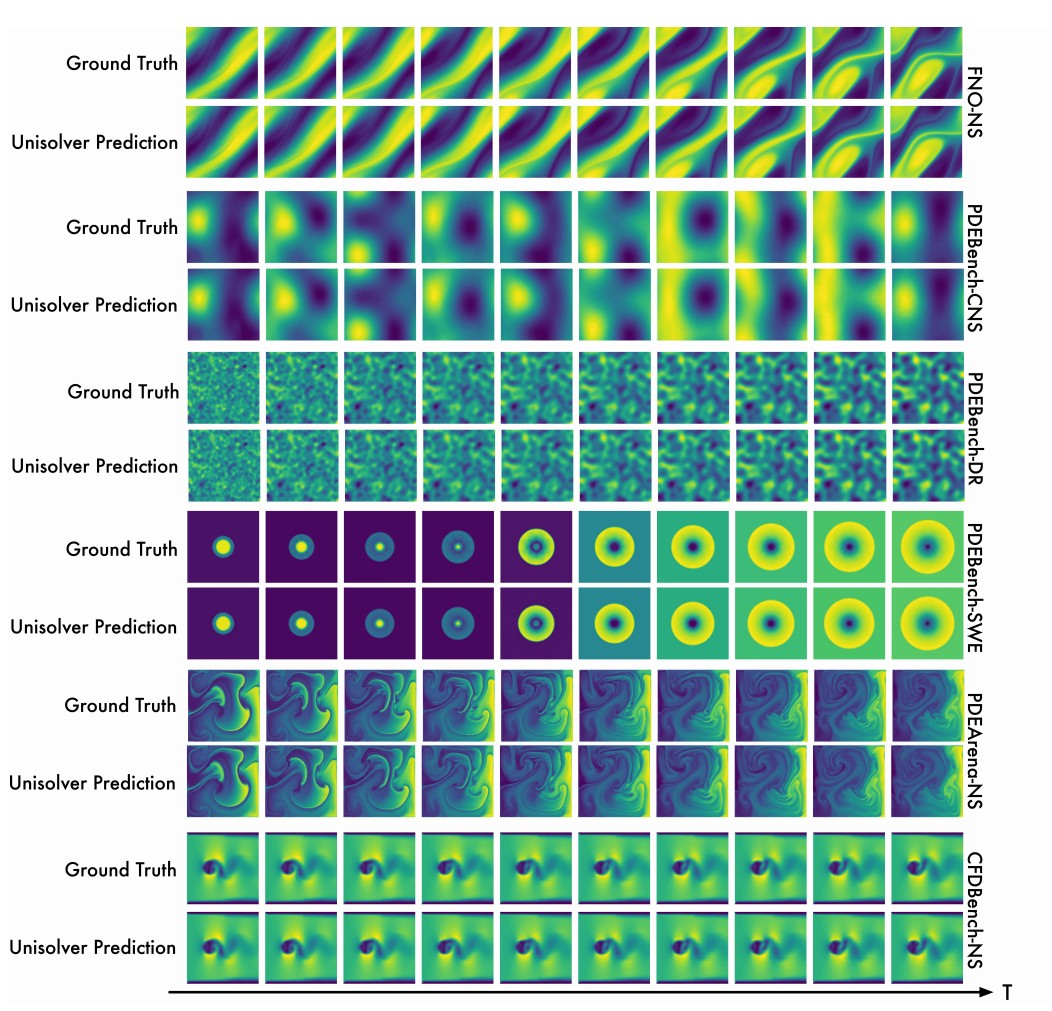

Figure 17: Visualization of the full trajectories in the 2D mixed PDEs, with the names of the subsets displayed on the right. Ground truth and Unisolver predictions are presented, visually highlighting the complexity and diversity of the 2D mixed PDEs.

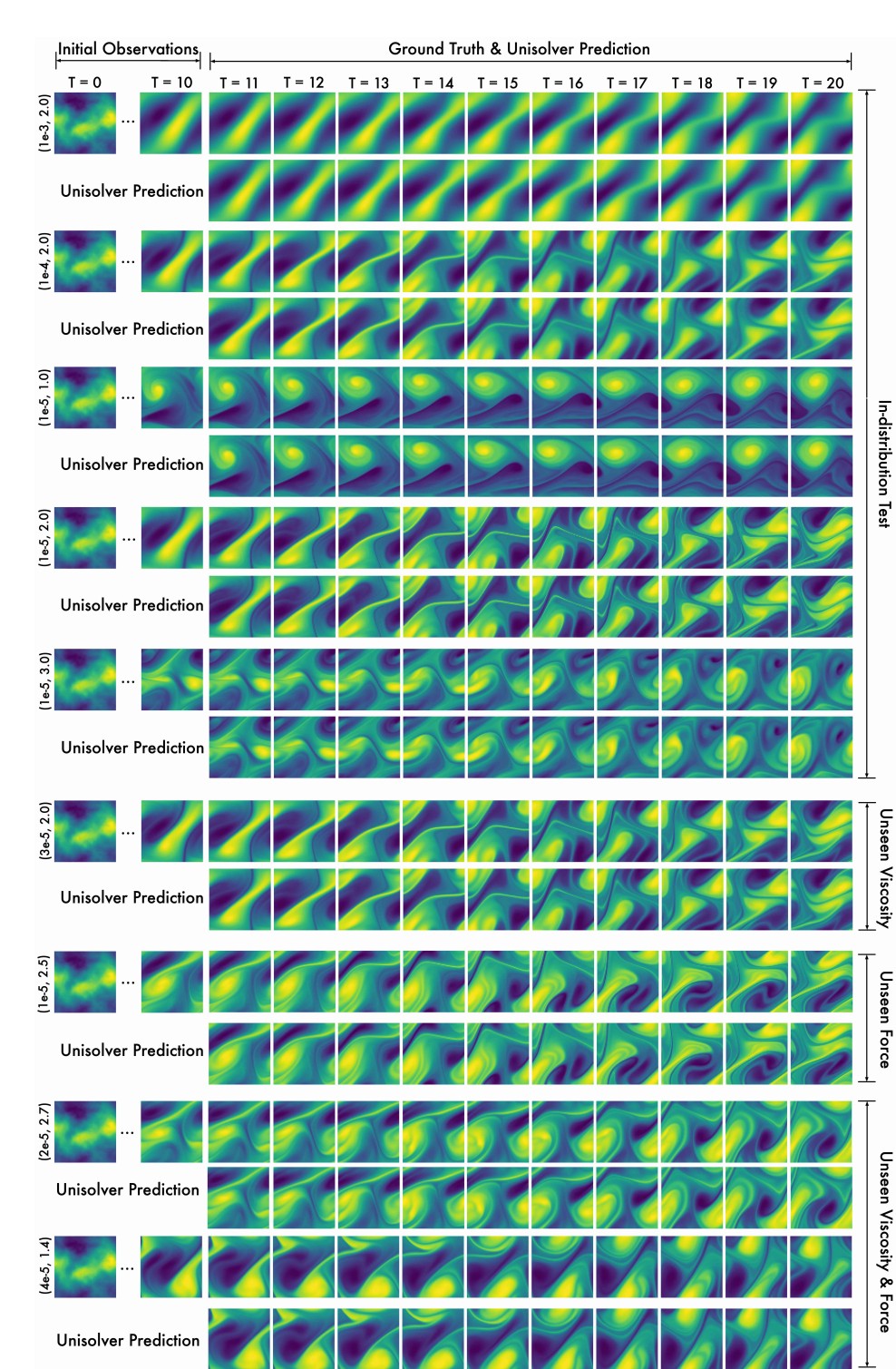

Figure 18: Visualization of the full trajectories in the HeterNS, where all trajectories share **the same initial condition** but differ in *viscosity* ($\nu$) and *force* ($\omega$) (shown beside each case by the pairs ($\nu, \omega$)).

## K  FULL PCA VISUALIZATION OF THE LLM EMBEDDINGS

Here we present the full visualization of the LLM embeddings for 1D PDEs. As stated in Appendix F.2, the 1D PDEs contain six coefficients as well as the source term and the viscosity term. For ease of view, the PCA visualization in Figure 7 contains PDEs with a zero source term and a zero viscosity term only. Figure 19 provides the full PCA visualization of the LLM embeddings containing varying source terms and viscosity terms. The coefficients can be zero or non-zero, and the source term and the viscosity term can be zero, a non-zero number or a function. Specifically, the equations sharing the same coefficients as the *Advection equation* but with varying source terms and viscosity terms are highlighted in dark green, showing that these similar equations are truly encoded into similar embeddings. The full visualization further illustrates how LLM embeddings retain the mathematical structure of the target PDE family. However, the orthogonal random vectors in Table 7 and the manually constructed encodings in Table 8 fail to maintain such a intricate mathematical structure.

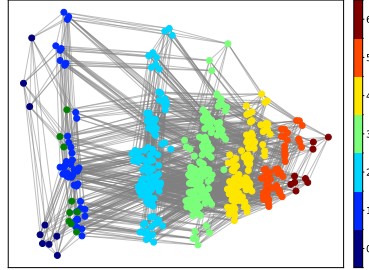

Figure 19: Full PCA visualization of LLM embeddings (varying source terms and viscosity terms). Different colors represent the number of non-zero coefficients, same as Figure 7. Further, we highlight the embeddings related to the Advection equation in dark green.

