# OpenReview forum: "Unisolver: PDE-Conditional Transformers Are Universal PDE Solvers"
_ICLR.cc/2025/Conference — Submitted to ICLR 2025_

### Official Review · Reviewer_fFn6 · 2024-10-30

**Soundness:** 4
**Presentation:** 4
**Contribution:** 3
**Rating:** 8
**Confidence:** 3

**Summary:**

This paper proposes a surrogate neural PDE solver model, called Unisolver, that adapts a transformer model with a novel conditioning mechanism for better generalization. Based on the analysis of a PDE-solving process, this work identifies complete PDE components, categorizes them into two groups (i.e., domain-wise and point-wise conditions), and proposes a decoupled conditioning.

**Strengths:**

This work promotes the research direction for developing a generalizable surrogate neural PDE solver. The proposed conditioning method is novel and clearly described with reasonable motivation and thorough analysis through experiments.

The experiments present a meaningful improvement of the proposed model in various examples, particularly highlighting the improved capacity for zero-shot generalization.

The ablation study validates the importance of the proposed ideas.

**Weaknesses:**

One of the essential requirements for having a reliable neural PDE solver is the capacity for accurate solutions for a long temporal evolution. The current experiments seemed to fix the temporal evolution up to 20 steps (shown in Fig. 17). (The accompanied videos show only 10 steps.) Experiments and analysis for long trajectory prediction will be invaluable.

Additionally, reproducibility will be improved if the codes are released. The proposed model relies on existing designs such as transformer, ViT, and DiT. This should not be a critical weakness of the work, though. Rather, It will be particularly useful to have actual codes because of the combinatory design of the proposed model.

**Questions:**

What will happen if the LLM model (i.e., conditioning equation symbols) is omitted in a single PDE dataset? I guess it may not affect the performance because such a dataset does not need to be conditioned on this component. For example, will this be true for the HeterNS example, which I guess that it embeds the same condition on equation symbols?

It looks like a good idea to embed the equation symbols using LLM; on the other hand, I am wondering if a simple manual encoding (e.g., representing each PDE term as a one-hot vector and composing them to represent a full PDE equation) could be enough for a similar performance. This question also relates to the "Random vector" ablation study in Table 7. How did the random vectors get assigned for each PDE?

---

> ### Author Response · Authors · 2024-11-20
> **Response to Reviewer fFn6 [Part 1]**
>
> We would like to sincerely thank Reviewer fFn6 for providing a detailed review and insightful questions.
>
> > **Q1:** "The current experiments seemed to fix the temporal evolution up to 20 steps. Experiments and analysis for long trajectory prediction will be invaluable."
>
> Thank you for this valuable suggestion. In our experiments, the 1D time-dependent PDEs benchmark indeed **includes 100 time steps** within the range [0, 1]. On this benchmark, Unisolver utilizes an implicit neural representation to decode the solution at any spatial-temporal coordinates within a fixed domain: $(x,t)\in[-1,1]\times[0,1]$.
>
> While on the HeterNS benchmark, the number of evolution steps is limited to 20, the actual time interval **between 2 consecutive frames is 1 second** which is a substantial temporal gap. A numerical solver would require approximately 10,000 time-stepping iterations to achieve precise one-second solutions.
>
> Further, as per your request, we extend the temporal evolution steps of HeterNS to 30 steps, corresponding to 30 seconds of complex fluid dynamics, and report the **zero-shot performance** comparison between Unisolver and the baselines in the table below. We present the performance on the subdataset with a viscosity coefficient of $\nu = 1 \times 10^{-5}$ and a force coefficient of $\omega = 2$. This is a particularly challenging task, as these models have never seen such long trajectories in the training data (at most 20 seconds). Despite this, Unisolver achieves the best performance. We have included this experiment in $\underline{\text{Appendix H.3 of the revised paper}}$.
>
> |             | Unisolver  | FNO    | ViT    | Factormer |
> | ----------- | ---------- | ------ | ------ | --------- |
> | Relative L2 | **0.1956** | 0.3105 | 0.2527 | 0.2962    |
>
> > **Q2:** "It will be particularly useful to have actual codes because of the combinatory design of the proposed model."
>
> We sincerely thank you for providing this helpful suggestion. To facilitate the understanding of Unisolver's design, we have included an example **inference code for the 1D time-dependent PDEs** benchmark in the $\underline{\text{supplementary material}}$, along with the **checkpoint of our largest Unisolver model**, which was trained for **3000 A100 hours on 3 million samples**. You can modify the PDE configurations like PDE coefficients, external forces and boundary conditions and inference with Unisolver by yourself. Furthermore, **we promise to make both training and inference code public upon acceptance.**
>
> > **Q3:** "What will happen if the LLM model (i.e., conditioning equation symbols) is omitted in a single PDE dataset?"
>
> Thank you for this insightful question. In the HeterNS benchmark, we do omit the equation symbol embeddings, as the entire dataset is based on a single PDE with varying coefficients and external forces. In this case, the equation symbol embeddings are unnecessary because all samples share the same underlying PDE structure. The LLM embeddings are helpful when the dataset contains multiple types of PDEs by **allowing the model to distinguish between different PDEs more effectively**. Consequently, only the viscosity coefficients and external forces are inputs to the model on the HeterNS dataset.

---

> ### Author Response · Authors · 2024-11-20
> **Response to Reviewer fFn6 [Part 2]**
>
> > **Q4:** "I am wondering if a simple manual encoding could be enough for a similar performance. This question also relates to the "Random vector" ablation study in Table 7. How did the random vectors get assigned for each PDE?"
>
> Sorry for the confusion. The baseline in $\underline{\text{Table 7}}$ using "random vector" encoding actually uses **orthogonal random vectors**. It is equivalent to a baseline using "one-hot" encoding to indicate which PDE is currently used. We have updated the description of this baseline $\underline{\text{in Lines 430 and 442 of the revised paper}}$.
>
> Moreover, following your request, we conduct additional ablation studies by manually constructing a one-hot vector for each PDE term and combining them to represent a full PDE. The results are shown in the table below. We observe that the LLM embedding case consistently outperforms the manual encoding case in both in-distribution tests and four zero-shot generalization settings, showing a **4.23% average improvement**. Although the manual encoding aims to preserve the mathematical structure of the PDE as much as possible, the handcrafted features **struggle to perfectly capture the mathematical structure provided by LLMs** visualized in $\underline{\text{Figure 7(b) of the original submission}}$, leading to a decrease in performance.
>
> Further, using an LLM to encode the symbolic information allows for **more flexible adaptation** to more diverse sets of PDEs. For example, when we want to extend the 1D PDE dataset to contain non-polynomial functions like sinusoidal terms. This flexibility is particularly advantageous, as it reduces the manual effort associated with designing unique embeddings for each new PDE component, enabling broader adaptability and scalability across various PDE applications. We have included this ablation study in $\underline{\text{Appendix B in the revised paper}}$.
>
> | Ablation Type        | In-distribution Test | Burgers $\nu=0.1$ | Burgers $\nu=0.01$ | Burgers $\nu=0.001$ | Advection $\beta=0.1$ |
> | -------------------- | -------------------- | ----------------- | ------------------ | ------------------- | --------------------- |
> | Manually constructed | 0.0282               | 0.0184            | 0.0675             | 0.1386              | 0.0679                |
> | **Ours**             | **0.0277**           | **0.0176**        | **0.0659**         | **0.1350**          | **0.0603**            |

---

> > ### Comment · Reviewer_fFn6 · 2024-11-22
> >
> > I appreciate the great effort of the authors to revise the paper and answer my questions.
> >
> > I would like to interpret the results of a new ablation study regarding my question Q4 from a slightly different angle. One of my main concerns was whether LLM is overkill to condition the equation symbols. Thus, I expected that a simple encoding might have a similar ability to provide a useful signal for the conditioning as long as it allows the model to distinguish between different PDEs. I recall that this is the main role of LLM in the proposed model. I would have been rather surprised if it showed a "significant" difference. Arguably, 4.23% doesn't seem to be the case in my view although it's still not crystal clear how this improvement comes.
> >
> > In summary, although the proposed model has room for further analysis and validation of each component as the discussion unfolds, I foresee the model will motivate the follow-up work sharing the same spirit. Thus, I will keep my positive rating of 8.

---

> > > ### Author Response · Authors · 2024-11-23
> > >
> > > Thank you for your insightful feedback. We would like to provide further clarification on the new ablation study.
> > >
> > > Firstly, we confirm that we strictly followed your request in Q4 to conduct the experiments.
> > >
> > > Secondly, as shown in $\underline{\text{Appendix B in the revised paper}}$, the promotion for the in-distribution test and the Burgers equation generalization is **relatively modest**, around 2.77%. However, the promotion for the Advection Equation generalization is interestingly **"significant"**. By analyzing the pretraining dataset of 50,000 samples used in this ablation, we find that only 68 samples share the exact manually constructed key as the Advection Equation, compared to 202 samples for the Burgers equation. This suggests that the manually constructed encodings may struggle to generalize to equations with limited presence in the training data.
> > >
> > > We believe the improved generalizability provided by the LLM embeddings stems not only from the capability of LLM embeddings  to distinguish between different PDEs but, **more importantly, from their capacity to encapsulate essential mathematical structures, enabling similar PDEs to help one another**.  To further illustrate this, we include a full PCA visualization of the LLM embeddings in $\underline{\text{Appendix K of the revised paper}}$​. The visualization demonstrates that equations derived from the Advection Equation with varying external forces or viscosity terms are mapped to closely related embeddings, which likely enhances generalization. In contrast, the manually constructed encodings may fail to capture the mathematical structures, as they use similar one-hot-like representations for different terms without considering their mathematical relationships.

---

### Official Review · Reviewer_awFj · 2024-10-31

**Soundness:** 2
**Presentation:** 3
**Contribution:** 2
**Rating:** 3
**Confidence:** 5

**Summary:**

This paper proposes the Universal PDE Solver (Unisolver) which is capable of solving a wide scope of PDEs by training a novel Transformer model on diverse data and conditioned on diverse PDEs. Unisolver defines a complete set of PDE components and flexibly embeds them as domain-wise (e.g. equation symbols) and point-wise (e.g. boundaries) conditions for Transformer PDE solvers. Unisolver has achieved consistent state-of-the-art results on three challenging large-scale benchmarks.

**Strengths:**

1. A novel and decoupled conditioning mechanism for introducing physics information into PDE solving.
2. Extensive experiments on three challenging large-scale benchmarks and state-of-the-art performance.
3. Easy to follow.
4. Substantial implementation details are included.

**Weaknesses:**

1. In the proof of Theorem 1 in Appendix D, the proof from line 942 to 971 and the first term in line 992 are just the d'Alembert solution of wave equations, which is well-known in PDE theory and physics and appeared in standard textbooks. In line 937, equation (9) should be  equation (6). It is suggested to remove these proofs and cite a textbook. Further, I think it is unneccessary to use Theorem 1 to introduce the concepts of domain-wise and point-wise PDE components, since these concepts are clear and easy to understand by themselves and the paragraph following Theorem 1 did not point out explicitly why coefficient a works domain-wisely and external force f point-wisely using the solution expression in equation (2), and some PDE components in table 1, such as equation symbols, did not appear explicitly in equation (2).
2. Using complete PDE information will certainly help the solution prediction. Regarding the motivation of the proposed method, please explain why you choose to predict solutions in a data-driven manner instead of using numerical solutions and PINNs, which both can take full advantage of complete PDE components. Especially, please disucss the pros and cons of Unisolver vs PINNs in terms of accuracy and computational efficiency.
3. Traditional operator learning method such as FNO train neural networks to generalize across different configuratons (such as viscosity coefficients and force terms) of the same type of PDE. Unisolver used large dataset to train a PDE-conditional Transformer model and expect it to generalize across diverse PDEs. Did different types of PDEs help each other during prediction? If possible, please give an ablation study to show the transfer learning capability of Unisolver among different types of PDEs, and justify the integration of a wide range of PDEs instead of treating them independently. The Burges and Convection equations, used in table 5 to show zero-shot generalization capability, are only special cases of the equations in line 1044 used for training. You could, for example, train Unisolver on Navier-Stocks equation and test it on the diffusion-reaction equation, or train and test Navier-Stocks equation on different domain geometry, to demonstrate the transfer learning capability of Unisolver among different types of PDEs.
4. Please explain why Unisolver works much better than PINO in table 3 in terms of relative error. In principle, by incorporating physics-informed loss, PINO is able to obtain accurate solutions.  Can we improve the accuracy of PINO if we weight the physics-informed loss more heavily?
5. Please give the computational cost to generate the datasets, including hardware used and simulation time, especially for HeterNS.

**Questions:**

Why use the LLM embedding of PDE symbol instead of Latex code? It is better to give an intuive example to show the addtional useful information provided by LLM beyond equation symbols.

---

> ### Author Response · Authors · 2024-11-20
> **Response to Reviewer awFj [Part 1]**
>
> We would like to sincerely thank Reviewer awFj for providing a detailed review and insightful questions.
>
> > **Q1:** "It is suggested to remove these proofs and cite a textbook. Further, I think it is unnecessary to use Theorem 1 to introduce the concepts of domain-wise and point-wise PDE components."
>
> Thanks for your valuable suggestion. As per your request, we have removed the proofs in the appendix and cited the proof from a textbook instead. Additionally, we have revised $\underline{\text{Section 3.1}}$ to use the vibrating string equation solely as a motivating example for introducing the concepts of domain-wise and point-wise PDE components. We believe that the $\underline{\text{revised paper}}$ delivers a more compact and direct description of our motivation.
>
> > **Q2:** "Please explain why you choose to predict solutions in a data-driven manner instead of using numerical solutions and PINNs, which both can take full advantage of complete PDE components. Especially, please disucss the pros and cons of Unisolver vs PINNs in terms of accuracy and computational efficiency."
>
> Firstly, as demonstrated in our experiments in "Incomplete component scenario" $\underline{\text{in Line 496 of Section 4.4}}$, Unisolver exhibits strong performance even **when some PDE components are incomplete**. In contrast, PINNs and traditional numerical methods are inapplicable under such circumstances.
>
> Further, regarding **accuracy**, it is well known that PINNs "often face severe difficulties and even fail to tackle problems whose solution exhibits highly nonlinear, multi-scale, or chaotic behavior"[1], and the optimization process of PINNs often faces thorny challenges [2], even for a simple steady Navier-Stokes Equation [3]. The datasets used in our work, such as HeterNS, can present significant challenges for PINNs due to their high non-linearity and chaotic nature.
>
> In terms of **computational efficiency**, as stated in $\underline{\text{Lines 35-36, 44-47 of main text}}$, PINNs and numerical methods also struggle to generalize to new PDEs with varying PDE components like coefficients or boundary conditions, requiring retraining for each new PDE. In contrast, although Unisolver requires some time for data collection and training, once the model is well trained, it can generalize to new PDEs without retraining and generate a solution within a second. The efficiency benefit is the most widely acknowledged advantage of Neural Operator than PINNs.
>
> Moreover, we provide the inference times and memory consumption for each model to predict a single frame on the HeterNS dataset, along with the calculation time and memory consumption for the numerical solver which is used to generate the dataset, as summarized in $\underline{\text{Appendix H.5 in the revised paper}}$. Unisolver is 1,000+ times faster than the numerical solver.
>
> [1] Respecting causality for training physics-informed neural networks. Computer Methods in Applied Mechanics and Engineering 2024.
>
> [2] RoPINN: Region Optimized Physics-Informed Neural Networks. NeurIPS 2024.
>
> [3] PINNsFormer: A Transformer-Based Framework For Physics-Informed Neural Networks, ICLR 2024
>
> > **Q3:** "Traditional operator learning method such as FNO train neural networks to generalize across different configuratons (such as viscosity coefficients and force terms) of the same type of PDE."
>
> We respectfully point out that **this comment exists some misunderstandings of FNO**. It is worth noting that in the original FNO paper, the model was not designed to handle PDEs with different configurations, but rather train separate models for **every single configuration**. Specifically, the original NS dataset proposed by FNO test the model's capability to **generalize to unseen initial conditions under a fixed PDE configuration**.
>
> In contrast, Unisolver handles a more general generalization task, requiring the model to generalize not only to unseen initial conditions, but also to other PDE components like the coefficients and external forces. The HeterNS benchmark is an extension of the original NS benchmark to test the models' capability to generalize across different configurations within the same PDE type, while the 1D time-dependent PDEs and 2D mixed PDEs benchmark require the model to generalize across different types of PDEs.
>
> To be more clear, we have listed the comparison between settings of Unisolver and FNO as follows and add $\underline{\text{Figure 2 in revised paper}}$ for clarity.
>
> |           | Training                                                     | Inference                                                    |
> | - | - | - |
> | FNO Paper | Train separate models for every configuration of PDE coefficient | Unseen initial conditions                                    |
> | Unisolver | Train one unified model for diverse PDE coefficients and external forces, etc. | Unseen initial conditions, viscosity and external forces, etc |

---

> ### Author Response · Authors · 2024-11-20
> **Response to Reviewer awFj [Part 2]**
>
> > **Q4:** "Did different types of PDEs help each other during prediction? If possible, please give an ablation study to show the transfer learning capability of Unisolver among different types of PDEs, and justify the integration of a wide range of PDEs instead of treating them independently."
>
> As per your request, we design an ablation study to evaluate the effect of joint training on the 1D time-dependent PDE benchmark. As stated in $\underline{\text{Appendix E of the original submission}}$, the general equation formulations used in this benchmark include two polynomials, $f_0$ and $f_1$, both with a maximum order of 3. We construct three distinct sub-datasets, each with 10,000 samples, to test the impact of joint training. The polynomials in each dataset are fixed to orders of 1, 2, and 3, respectively, ensuring that the PDEs contained in these three datasets do not overlap. For instance, in the dataset with polynomials of order 3, only $c_{03}$ and $c_{13}$ are non-zero terms, while $c_{01}$, $c_{02}$, $c_{11}$ and $c_{12}$ are fixed to zero.
>
> We conduct both joint training and independent training for 500 epochs on these 3 subdatasets. The results are shown in the table below.
>
> | Polynomial order     | 1      | 2      | 3      |
> | -------------------- | ------ | ------ | ------ |
> | Independent Training | 0.0792 | 0.1161 | 0.1236 |
> | Joint Training       | 0.0555 | 0.0738 | 0.0695 |
>
> It can be shown that joint training significantly boosts performance on three datasets, even though the 3 types of PDEs do not overlap, demonstrating that these PDEs do "help each other" during training. We have included this experiment in $\underline{\text{Appendix C in the revised paper}}$.
>
> As for the transfer learning capability of Unisolver, we have provided a equation generalization experiment in $\underline{\text{Appendix G.3 of original submission}}$. The results demonstrate that Unisolver has strong generalization capabilities to unseen PDEs, even when the downstream PDEs are entirely new.
>
> > **Q5:** "Please explain why Unisolver works much better than PINO in table 3 in terms of relative error. Can we improve the accuracy of PINO if we weight the physics-informed loss more heavily?"
>
> Thanks for pointing this out. Firstly, we would like to detail that PINO in Table 3 is essentially a FNO model augmented with a physics-informed loss, as we stated in $\underline{\text{Lines 319-321 of main text}}$. It is worth noting that although physics-informed loss proposed by PINO seems to help the model training, in pratice, **this loss cannot be accurately calculated in experimented data due to the finite spatiotemporal resolution.**
>
> The HeterNS benchmark, as well as the original NS benchmark in the FNO paper, employs a coarse spatial resolution of $64 \times 64$ and a long temporal interval of 1 second. Meanwhile, the official implementation of PINO utilizes a finite difference method to compute the temporal derivatives, which cannot produce an accurate estimation of derivatives given the long temporal interval between two consecutive solution frames.
>
> We try to raise the weight of the physics-informed loss for PINO, and report the results below. The results show that the performance of PINO gets worse if we raise the weight of the physics-informed loss.
>
> | Viscosity $\nu $  | 1e-5   | 5e-5   | 1e-4   | 5e-4   | 1e-3   | 8e-6   | 3e-5   | 8e-5   | 3e-4   | 8e-4   | 2e-3   |
> | ----------------- | ------ | ------ | ------ | ------ | ------ | ------ | ------ | ------ | ------ | ------ | ------ |
> | PINO (weight=0.1) | 0.1012 | 0.0443 | 0.0263 | 0.0073 | 0.0031 | 0.1014 | 0.0646 | 0.0299 | 0.0142 | 0.0081 | 0.1894 |
> | PINO (weight=1)   | 0.1954 | 0.0985 | 0.0600 | 0.0093 | 0.0036 | 0.1278 | 0.1418 | 0.0578 | 0.0177 | 0.0107 | 0.1962 |
>
> | Force $\omega$    | 1      | 2      | 3      | 0.5    | 1.5    | 2.5    | 3.5    |
> | ----------------- | ------ | ------ | ------ | ------ | ------ | ------ | ------ |
> | PINO (weight=0.1) | 0.0914 | 0.1012 | 0.2707 | 1.0570 | 0.5010 | 0.4660 | 0.8380 |
> | PINO (weight=1)   | 0.1167 | 0.1954 | 0.2943 | 0.8662 | 0.4549 | 0.4383 | 0.7406 |

---

> ### Author Response · Authors · 2024-11-20
> **Response to Reviewer awFj [Part 3]**
>
> > **Q6:** "Please give the computational cost to generate the datasets, including hardware used and simulation time, especially for HeterNS."
>
> Note that the 1D time-dependent PDEs and 2D mixed PDEs benchmark are not proposed by us and we cannot find the exact data generation time in public resources. However, we contacted the authors of PDEformer by email, and they said that it took about 2 weeks to generate "1D time-dependent PDEs" with 96 CPU cores. As for the 2D mixed PDEs benchmark, it is collected by DPOT with several data sources, which is hard to calculate the extract generation time due to inconsistent experiment environments.
>
> As for the HeterNS benchmark, we use a single A100 GPU to generate all the training and test samples. The total simulation time is about 35 hours for generating the 15000 training samples. Again, we want to highlight that the inference efficiency of Unisolver should not be underestimated, which does not require retraining for new tasks, leading to an efficient surrogate model for traditional numerical methods.
>
> > **Q7:** "Why use the LLM embedding of PDE symbol instead of Latex code? It is better to give an intuitive example to show the additional useful information provided by LLM beyond equation symbols."
>
> Unlike numerical information such as the PDE coefficients and external forces, the LaTex codes of the PDE symbols are essential strings that cannot be feed into the neural network directly. Therefore, we use an LLM to encode the LaTeX codes into a fixed-dimensional vector.
>
> Meanwhile, $\underline{\text{Figure 6(b)}}$ shows the PCA visualization of the LLM embeddings. We can observe that the LLM is able to encode PDEs with similar complexities into similar embeddings, which intuitively shows that the equation symbol embeddings capture prior mathematical information beyond the literal symbols.
>
>
>
> Thank you again for your thoughtful feedback. We noticed that you have rated our paper with a score of 3, and assigned a confidence level of 5, which indicates a rather negative evaluation. If you have any further questions or require additional clarification, we would be pleased to offer additional explanations.

---

> > ### Comment · Reviewer_awFj · 2024-11-25
> > **Feedback to the authors' responses**
> >
> > Thank you very much for your very detailed responses. Most of my concerns have been addressed. It seems that training a transformer conditioned on all PDE components, using large datasets of diverse types of PDEs, really improves the prediction accuracy.  It has been shown that joint training boosts performance on synthetic datasets and improves the generalization capabilities to unseen PDEs on real datasets. The motivating example has been rewritten and is much reasonable now. However, because in the original version, the proof of the d'Alembert solution for wave equations was described without a citation, I decided to maintain my score.

---

> > > ### Author Response · Authors · 2024-11-25
> > >
> > > Thank you for your feedback.
> > >
> > > First, we would like to emphasize that **we have cited the book *Fourier Analysis: An Introduction*** in $\underline{\text{Line 169 of the original submission}}$ when introducing the vibrating string equation, which contains the proof for its solution. For your reference, here is an openly accessible link to the book: http://kryakin.site/am2/Stein-Shakarchi-1-Fourier_Analysis.pdf, where the proof can be found on Pages 8-11.
> > >
> > > Second, we sincerely apologize for any misunderstanding. **We would like to clarify that we have never claimed the proof of the vibrating string equation's solution as our original contribution, nor have we suggested that we were the first to prove this theorem.** The proof was included in the original submission purely for completeness and to help readers better understand the motivating example.
> > >
> > > Third, in the revised paper, we have removed the proof and explicitly cited Evans’s *Partial Differential Equations* as the reference for a detailed proof.
> > >
> > > We hope this further clarification adequately addresses your concerns.

---

> > > > ### Comment · Reviewer_awFj · 2024-11-28
> > > > **Response to Official Comment by Authors**
> > > >
> > > > Thanks for your clarification. Regarding the citation, the standard way is to cite the book right above the proof and state explicitly "included for completeness". Besides, the mathematics in original Appendix D is described in a casual way. For example, you started a "proof", but even did not formally and explicitly describe what you were going to prove (e.g. using a lemma or proposition). There are some other unstrict places. The theoretical part of the original version is actually reluctant and inaccurate. I did not point out such problems at this first time simply because I suggest to remove them due to the citation issue. I help you to improve the paper, but I will maintain my judgement.

---

> > > > > ### Author Response · Authors · 2024-11-28
> > > > > **We respect your right of rating but your decision might violate the ICLR Reviewer Guidelines.**
> > > > >
> > > > > Thanks for your reply and the pursuit of scientific rigor, which we highly respect. We also sincerely appreciate your help in reviewing and improving our paper.
> > > > >
> > > > > According to your response, **it is clear that the only reason that you rate 3 for our paper is about the proof part of our original version, which has been completely revised in our current paper.**
> > > > >
> > > > > We respect your right to rate our paper since you are the reviewer and greatly help us with your suggestions. However, again, we have to highlight the **ICLR Reviewer Guidelines** (https://iclr.cc/Conferences/2024/ReviewerGuide). According to the **"Update your review, taking into account ... any revisions to the submission"** rule, we kindly remind you that your decision might violate the ICLR Reviewer Guidelines.
> > > > >
> > > > > Since **your rating is not about the current quality of our revised paper** and it seems impossible for you to change your score, it seems that the discussion has come to a deadlock.
> > > > >
> > > > > Sincerely thanks for your time and dedication in reviewing our paper.
> > > > >
> > > > > Best regards.

---

> > > ### Author Response · Authors · 2024-11-27
> > > **We are anticipating your response. Hope the reviewer can consider our revised paper.**
> > >
> > > Dear Reviewer awFj,
> > >
> > > Sincerely thanks for your dedication to our paper. Your remarks on verifying if diverse PDEs can help model performance are quite valuable. Also, your suggestions for our writing are very rigorous and constructive.
> > >
> > > In your last response, you mentioned that the only reason for rejection is that "in original version, we do not include a citation in our proof". As we clarified in the last response, we have included a citation for the textbook in the $\underline{\text{Line 169 of main text of the original submission}}$ and have never attempted to claim the proof for motivating example as our contribution. As we listed in $\underline{\text{Line 96-97 in main text}}$, the only point we want to highlight is how this basic PDE problem motivate us to design Unisolver.
> > >
> > > We are truly sorry for the possible misleading in our previous version and strictly followed your suggestion and rephrased all the corresponding parts in the $\underline{\text{revised paper}}$. As highlighted in the **ICLR Reviewer Guidelines** (https://iclr.cc/Conferences/2024/ReviewerGuide), the final recommendation requests the reviewer "**Update your review, taking into account ... any revisions to the submission.**" Thus, we sincerely hope you can consider our revised paper for your final conclusion.
> > >
> > > In Unisolver, we have provided extensive experiments, ablations and visualizations, leading to a paper with **more than 30 pages**. As acknowledged **in your original review**, our method is **"novel" and "easy to follow"** method with **"extensive experiments"** and **"substantial implementation details"**. We believe that this paper can be a good supplement to current research. It would indeed be regrettable if this work were to be rejected solely due to a citation oversight in a previous revision.
> > >
> > > Thanks again for your review. Looking forward to your response.

---

> ### Author Response · Authors · 2024-11-25
> **Request of Reviewer’s attention and feedback**
>
> Dear Reviewer awFj,
>
> We kindly remind you that there are only **two days left** of the two-week Reviewer-author discussion period. So please kindly let us know if our response has addressed your concerns. We will be happy to deal with any further issues/questions.
>
> We made every effort to address the concerns you suggested:
>
> - The vibrating string equation in $\underline{\text{Section 3.1}}$ is now used solely as a motivating example to introduce the concepts of domain-wise and point-wise PDE components. We also remove the proof in $\underline{\text{Appendix E}}$ and cite standard references.
> - We elaborate on the advantages of Unisolver over PINNs and numerical solvers, especially in terms of **generalizability and computational efficiency**. We also include inference time and memory consumption comparisons between Unisolver and numerical solvers in $\underline{\text{Appendix H.5 of the revised paper}}$.
> - We add an ablation study on the 1D time-dependent PDE benchmark in $\underline{\text{Appendix C.1}}$ to demonstrate that joint training on different PDEs significantly improves performance. We also provide Unisolver's capability to generalize to entirely new PDEs in $\underline{\text{Appendix C.2}}$.
> - We provide detailed explanations on PINO's performance and add an experiment that raises the weight of the physics-informed loss following your suggestion.
> - We provide detailed generation costs of the HeterNS dataset.
> - We explain the reason of using an LLM to encode the symbolic information of PDEs.
>
> These updates have already been incorporated into the $\underline{\text{revised paper}}$. Inference code is also included in $\underline{\text{supplementary material}}$.
>
> We hope our responses and revisions have addressed all your concerns effectively.
>
> Thanks again for your dedication to reviewing our paper. Looking forward to your reply.

---

### Official Review · Reviewer_1Nnh · 2024-11-01

**Soundness:** 3
**Presentation:** 4
**Contribution:** 3
**Rating:** 8
**Confidence:** 3

**Summary:**

The paper proposes a new foundation model for solving PDEs. The core contribution is a systematic way to include all possible PDE information in the model.  The PDE information is divided into point-wise and domain-wise information, such as the PDE parameters. Most notably, the authors propose to include the mathematical equation of the PDE into the model by using an LLM encoding. The model is evaluated on three large-scale benchmarks and is shown to surpass the considered baselines.

**Strengths:**

1. Good experimental design, including challenging tasks and an appropriate choice of baselines. The authors provided an evaluation of the standard deviation in the appendix as well.
2. Stronger performance than comparable foundation model architectures.
3. Simple but powerful architectural design considering all aspects encountered while solving PDEs.
4. The paper is clearly written and easy to follow.

**Weaknesses:**

1. The components defining the behavior of a PDE are already known and not a "key finding," as stated in the abstract.
2. Limited methodological novelty (but a good combination of existing components).
3. No evaluation of inference times and memory consumption compared to the baselines; only the training time and parameter count are provided.
4. The claim that the LLM uses its mathematical knowledge obtained from pretraining on language data is not well supported. The ablation shows that the performance drops when not using the LLM by comparing it to a random vector encoding, but it is not shown that this is due to prior mathematical knowledge of the LLM. Another explanation for the good results of using the LLM encodings could be that they are simply a good representation of the number of symbols in the equation or that they make it easy for the model to differentiate between the considered PDEs.

**Questions:**

1. Regarding the LLM ablations:
    1. Please clarify how you performed the random vector encoding. Does random vector encoding mean that all the different equation combinations get their own fixed random vector? Or is the model fed with a new, different random vector for every input?
    2. Wouldn't a one-hot encoding of which PDE version is currently used be a better baseline?
    3. Another possible experiment could be to train the model with “wrong” mathematical encodings. For example, map “u_{xx}” to “b_{vvv}” or a “+” to “-”  (with fixed maps and keeping the number of elements in the equation identical). This would keep the string encoding close, but if the LLM really used mathematical understanding, this should lead to a worse performance.
2. In line 1347, you introduce a binary mask to notice the model which component is active. Shouldn’t the LLM encoding of the equation handle this?
3. How costly is using an LLM to encode the equation?
4. Is the model trained with one-step predictions or autoregressive rollouts?
5. How are the inputs aggregated in the “deep condition consolidation”?
6. Please clarify in the paper on all figures that the numbers refer to the relative L2 (missing in Fig. 8, Fig 6a, Tab7.), especially since you switch between relative L2 and relative L2 x 10^-2.

---

> ### Author Response · Authors · 2024-11-20
> **Response to Reviewer 1Nnh [Part 1]**
>
> We would like to sincerely thank Reviewer 1Nnh for providing a detailed review and insightful questions.
>
> > **Q1:** "The components defining the behavior of a PDE are already known and not a "key finding," as stated in the abstract."
>
> Many thanks for your feedback. We agree that the components defining the behavior of a PDE are already known. We have removed the proofs in the appendix and revised the $\underline{\text{Abstract and Section 3.1}}$ to use the vibrating string equation solely as a motivating example. We believe that the $\underline{\text{revised paper}}$ delivers a more compact and clear description of our motivation.
>
> > **Q2:** "Limited methodological novelty (but a good combination of existing components)."
>
> Thanks for your appreciation of our overall design. It is worth noticing that while our approach leverages existing components, the design is non-trivial. We carefully consider each component and develop tailored embedding strategies to capture their unique characteristics effectively.
>
> Furthermore, we are the first to clearly define a method for integrating PDE information into deep models via conditional modeling. This comprehensive approach leads to highly effective performance as demonstrated by the extensive experimental results.
>
> > **Q3:** "No evaluation of inference times and memory consumption compared to the baselines."
>
> As per your request, we provide the inference times and memory consumption for each model to predict a single frame on the HeterNS dataset, along with the calculation time and memory consumption for the numerical solver which is used to generate the dataset to calculate the next frame, as summarized in the table below. The results are measured on an A100 GPU with a batch size of 1. Unisolver demonstrates **comparable inference speed** to FNO, while consuming **less memory**. Besides, all neural PDE solvers are approximately 1,000 times faster than the numerical solver. We have included the computational cost comparison in $\underline{\text{Appendix H.5 of the revised paper}}$.
>
> |                  | Average Inference (Calculation) Time/s | Average Memory Usage/MB |
> | ---------------- | -------------------------------------- | ----------------------- |
> | FNO              | 0.0042                                 | 730                     |
> | PINO             | 0.0042                                 | 730                     |
> | ViT              | 0.0045                                 | 558                     |
> | Factformer       | 0.0103                                 | 758                     |
> | ICON             | 0.0057                                 | 784                     |
> | MPP              | 0.0120                                 | 1200                    |
> | Unisolver        | 0.0054                                 | 554                     |
> | Numerical Solver | 7                                      | 524                     |
>
> > **Q4:** "The ablation shows that the performance drops when not using the LLM by comparing it to a random vector encoding, but it is not shown that this is due to prior mathematical knowledge of the LLM."
>
> We appreciate your thoughtful feedback. To further verify the role of LLM embeddings in encoding PDE information, we conduct two additional ablation experiments. In the first experiment, the LLM only encodes **the number of non-zero terms** in the 1D PDE. In the second experiment, the LLM encodes the **"wrong" PDE information**. Specifically, we replace “*” with “/” and adjust polynomial orders to their reciprocals. For example, the original latex code $u\_t + c\_{01} * u + c\_{02} * u^2 + s(x) + (c\_{11} * u + c\_{13} * u^3)\_x = 0$ is transformed into $u\_t + c\_{01} / u + c\_{02} / u^{1/2} + s(x) + (c\_{11} / u + c\_{13} / u^{1/3})\_x = 0$. The results of these two ablation studies are shown below.
>
> | Abaltion type                        | In-distribution Test | Burgers $\nu=0.1$ | Burgers $\nu=0.01$ | Burgers $\nu=0.001$ | Advection $\beta=0.1$ |
> | ------------------------------------ | -------------------- | ----------------- | ------------------ | ------------------- | --------------------- |
> | Encode Num of non-zero terms         | 0.0285               | 0.0180            | 0.0665             | 0.1391              | 0.0618                |
> | Encode "wrong" mathematical encoding | 0.0289               | 0.0181            | 0.0672             | 0.1361              | 0.0619                |
> | **Ours**                             | **0.0277**           | **0.0176**        | **0.0659**         | **0.1350**          | **0.0603**            |

---

> > ### Comment · Reviewer_1Nnh · 2024-11-23
> >
> > Thank you for addressing my concerns and questions. I have raised my score to 8. In general, I believe that the provided model and datasets present a valuable contribution. I think it is a really interesting result that the performance of the LLM encodings doesn't seem to be matched by manually constructed ones, even in the new ablations.  The previous claims of novel findings regarding the general understanding of PDEs were too far-reaching, in my opinion, and distracting from the main contributions. Removing them has improved the paper.
> >
> > Regarding **Q1:**  Thank you for updating the abstract. However, the conclusion still states the similar claim
> >
> > > Concretely, Unisolver proposes a complete set of PDE components
> >
> > in Line 936.

---

> > > ### Author Response · Authors · 2024-11-24
> > >
> > > Thank you for your response and raising the score. We greatly appreciate your further suggestions regarding the conclusion and have revised the corresponding claims in the latest version of the revised paper.

---

> ### Author Response · Authors · 2024-11-20
> **Response to Reviewer 1Nnh [Part 2]**
>
> The results indicate that the model indeed obtains additional information beyond merely the count of non-zero terms from the LLM embeddings. Moreover, embedding "wrong" mathematical information generally leads to a decline in performance, highlighting the importance of accurately embedding the PDE information. While we cannot definitely claim that the LLM "understands" mathematical knowledge, we can confirm that the use of LLM enables us to encode useful mathematical information into deep representations. Furthermore, we also update the corresponding "mathematical knowledge" to "mathematical information" in the $\underline{\text{revised paper}}$ for scientific rigor. Meanwhile, the additional ablation studies have also been included in $\underline{\text{Appendix B of the revised paper}}$.
>
> > **Q5:** "Please clarify how you performed the random vector encoding."
> >
> > Wouldn't a one-hot encoding of which PDE version is currently used be a better baseline?
>
> Sorry for the confusion. The baseline using "random vector" encoding actually uses **orthogonal random vectors**. It is equivalent to a baseline using "one-hot" encoding to indicate which PDE is currently used. We have updated the description of random vector to orthogonal random vectors in $\underline{\text{Line 430 and 442 in the revised paper}}$ accordingly.
>
> > **Q6:** "You introduce a binary mask to notice the model which component is active. Shouldn’t the LLM encoding of the equation handle this?"
>
> Thank you for your thoughtful feedback. In the data pre-processing stage, if a component does not exist in a certain dataset, we set this component to zero by default. However, even a zero-valued component might also hold its physical significance—for example, a zero-valued external force may convey specific physical meaning. Therefore, we include an additional binary mask to indicate to the model whether a component is an actual PDE term or simply set to zero by default. While the LLM embedding can provide some indication of this information, it does not serve as the input to the encoders of other components. This binary mask, however, aids the encoder's learning and further clarifies the information without introducing significant computational overhead. Moreover, we update the description in $\underline{\text{Line 240 in the revised paper}}$.
>
> > **Q7:** "How costly is using an LLM to encode the equation?"
>
> Thank you for your insightful question regarding the computational overhead of generating LLM embeddings. It takes about 450 seconds to get the embeddings for all 576 types of PDEs using the **Llama3-8B** model on a single A100 GPU.
>
> Note that large language models have been extensively optimized for engineering applications by extensive researchers and companies, which allows for efficient processing within reasonable time constraints.
>
> > **Q8:** "Is the model trained with one-step predictions or autoregressive rollouts?"
>
> For the HeterNS and the 2D mixed PDEs benchmark, following previous methods FNO and DPOT, we train the model using one-step predictions and test the model in an autoregressive manner. For the 1D time-dependent PDEs benchmark, we train the model to predict the solution at specific spatial-temporal coordinates through an INR following PDEformer.
>
> Moreover, we update the description in $\underline{\text{Line 186, 214 and 230 in the revised paper}}$.
>
> > **Q9:** "How are the inputs aggregated in the “deep condition consolidation”?"
>
> Sorry for the confusion. In this paper, we simply add the deep conditions within the same class. And we have updated the description in $\underline{\text{Line 256 in the revised paper}}$.
>
> > **Q10:** "Please clarify in the paper on all figures that the numbers refer to the relative L2, especially since you switch between relative L2 and relative L2 x 10^-2."
>
> Thank you for pointing this out. We have updated the corresponding figures and tables to clearly indicate the metrics we use $\underline{\text{in the revised paper}}$.

---

### Official Review · Reviewer_uAAv · 2024-11-02

**Soundness:** 1
**Presentation:** 1
**Contribution:** 2
**Rating:** 3
**Confidence:** 3

**Summary:**

In this paper they add symbolic representation of the PDE operator to an LLM based PDE solver.  For the one dimensional wave equation, they add Prompt: "u_{tt} - aˆ2 u_{xx} = f(x,t)"".  More PDE prompts are included in Table 12 of the Appendix. The model is trained "for solving multitudinous PDEs universally."   The architecture is based on Theorem 1, which shows that the PDE solution depends on the PDE equation, the input data, the forces, and the boundary data.   The method is compared to several baselines. However,  since the baselines do not include the language model prompt for the PDE they augment these baselines "by providing sufficient physics information to ensure a fair comparison, either by concatenating the inputs with varied PDE components, providing prompting trajectories (ICON) or applying physics-informed loss (PINO)."

**Strengths:**

The strength of the paper is the implementation of an LLM based PDE solvers, which includes prompt describing the PDE equation, e.g. Prompt: "u_{tt} - aˆ2 u_{xx} = f(x,t)"".
They introduce HeterNS, an extension of the widely used 2D NS dataset (Li et al., 2021a), to assess how models handle diverse PDE components, particularly viscosity coefficients and force terms.
They also perform experiments on 1d time dependent PDEs, and 2d mixed PDEs.  They claims results which mostly beat the performance of their model, compared to the modified baselines.

**Weaknesses:**

PDE solvers are a machine learning approach to a problem where, if the inputs are properly specified, and the PDEs is well-posed, there is no need for machine learning: standard scientific computing results give very precise and well-understood solvers. So machine learning approached need to be properly motivated, and the problem needs to be carefully defined.

In this paper, the problem is not clearly defined.  I understood that an LLM method was used, along with a text prompt including the equation for the PDE.  But it was not clear exactly what the PDE was trained on, in for example, Table 2.  The training times are extremely long, compared to standard methods.   Also the modification to the baselines is not clearly described.  Inputs, outputs, and data need to be specified.
There are not standard datasets with standard performance measures: these are new datasets defined by the authors, and performance measures and baselines defined by the authors.  So the meaning and importance of the numbers in the table is really not clear.
To repeat: it's not clear to me what the models were trained on exactly, and it's not clear how the errors were measured, or what the meaning of the experiments are.  See questions below.

Even when the problem is underspecified, or the PDE is not well posed, there are a number of approaches which seek to find solutions.  For example, there is a notion of weak solutions to PDEs which lack classical solutions.  https://en.wikipedia.org/wiki/Weak_solution There is the notion of inverse problems, for learning operators from data. https://en.wikipedia.org/wiki/Inverse_problem . There is also the notion of homogenization, which deals with small scales in the data. https://en.wikipedia.org/wiki/Asymptotic_homogenization .

This paper critiques existing methods as not taking full advantage of of the data, but this is a misrepresentation of the learning problem for previous methods, which are expressly designed to solve the PDE without knowing the form of the equation.


Contribution: "Motivated by the mathematical structure of PDEs, we define the concept of complete PDE components, classify them into domain-wise and point-wise categories, and derive a decoupled conditioning mechanism for introducing physics information into PDE solving."
- This is a concept completely original to the paper, which ignores exiting expertise on PDE operators, and is not clearly explained.  It's also not clear how it corresponds to what they do in the paper.

Theorem 1: This is not a theorem. It is a restatement of an undergraduate textbook example of the solution of the analytical solutions to the one dimensional wave equation.  But it is not clear how this solution formula relates to a universal solution method for PDEs.  The solution expressed in the theorem is particular to the equation, and there do not exist similar explicit solutions for most PDEs.  For example, the heat equation solution depends in a very different way on the initial data: in a special case, it is simply convolution with a Gaussian.  The analogy no longer holds.

"In this paper, we will elaborate on how Unisolver finely and completely embeds all considered PDE components (Table 1) into deep PDE conditions based on our insights from the mathematical analysis."
- all the equation prompts had constant coefficients.  How would you prompt a two dimensional equation, with variable coefficients?
- How do you distinguish between the exact  PDE and incomplete knowledge of the PDE?

"Unisolver stems from the theoretical analysis of the PDE-solving process."
- How so? There is no universal PDE solver, since PDEs do not universally have solutions.
- Some PDE operators are well-posed, which means there existing unique solutions which are stable under certain types of perturbations of the data..  Others are not:  For example the wave equation with have multiple solutions if the initial data is mis-specified.  Some PDEs develop finite time singularites.
- How is the question of PDE ill posed / well posed addressed by this paper?

Paragraph 2: "previous neural solvers can be broadly categorized into two paradigms:  physics-informed neural networks (PINNs) (Raissi et al., 2019) and neural operators"
- This is an imprecise characterization.  There are multiple formulations of PDEs problems: these include: (i) learning a solution operator, based on a dataset of solutions on grids, and (ii) learning a single PDEs solution, based on incomplete data, as well as knowledge of the PDEs, as well as (iii) learning the PDE operator from sample(s) of (possibly incomplete data.

**Questions:**

"We compare Unisolver with six advanced baselines on the HeterNS to demonstrate its generalizability under varied PDE components: the well-established FNO (2021a), PINO (2021b) and ViT (2020) and current state-of-the-art methods Factformer (2023b), ICON (2023) and MPP (2023). We augment these baselines by providing sufficient physics information to ensure a fair comparison, either by concatenating the inputs with varied PDE components, providing prompting trajectories (ICON) or applying physics-informed loss (PINO). Furthermore, we compare Unisolver with two generalizable solvers—PDEformer (2024) and DPOT (2024) on zero-shot generalization performance in a head-to-head manner. We refrain from including additional baselines on these two benchmarks due to the substantial computational cost of using million-scale samples."
- what exactly is the training data, expressed in terms of the solutions to which PDEs?  Was it all the PDE solutions, were the text prompts included with the data?  What is the test data?  An initial condition? Along with the text of the PDE?
- Please describe what were the inputs to the training dataset, what were the inputs to the model (besides the latex text prompt for the PDE) and he models trained on a dataset of PDEs for each of the three main cases?
- what were the millions-scale samples?

This paper critiques existing methods as not taking full advantage of of the data, but this is a misrepresentation of the learning problem for previous methods, which are expressly designed to solve the PDE without knowing the form of the equation.    On the other hand, if the LLM is given the form of the PDE (as a text prompt), what is stopping it from simply using an existing scientific computing method solver to solve the PDE precisely?  Or is this meant to be the point of the method?  The lack of problem definition makes it hard to asses the contribution.

Contribution 1: Unisolver achieves consistent state-of-the-art performances across three large-scale benchmarks with impressive relative gains and presents favorable generalizability and scalability.
- what is meant by "favorable generalizability"?

"Our key finding is that a PDE solution is fundamentally under the control of a series of PDE components, e.g. equation symbols, coefficients, and boundary conditions."
- this key finding is not correct, and ignores many years of expertise in PDEs.  Please clarify and explain more precisely how this motivated your architecture.

"Thus, due to the omission of PDE information, current neural operators are mainly trained and tested on a limited set of PDEs."
  - This is imprecise, and misleading.  It suggests that the omission of the PDE operator is a flaw in current neural operator methods.  But that is not the case: these methods are designed to learn the PDE solution operator from data.  If we included knowledge of the PDE equation, we could simply avoid machine learning and solve the PDE using traditional scientific computing methods.
- Please explain how this method compares with and related to traditional scientific computing solvers.  Was the method trained on code for these solvers?

"We augment these baselines by providing sufficient physics information to ensure a fair comparison, either by concatenating the inputs with varied PDE components, providing prompting trajectories (ICON) or applying physics-informed loss (PINO)."
- how is this done?  These are no longer baselines, but new methods.

"Concretely, Unisolver proposes a complete set of PDE components and embeds these components into domain-wise and point-wise deep conditions separately for deep conditions. "
- This is not clear: what are "point-wise deep conditions "

---

> ### Author Response · Authors · 2024-11-20
> **Response to Reviewer uAAv [Part 1]**
>
> We sincerely thank Reviewer uAAv for providing a detailed review and insightful questions.
>
> > **Clarify some misunderstandings in the review.**
>
> First, we would like to clarify a few points that may have led to some misunderstandings to ensure that our work is accurately represented.
>
> **(1) About Method: Unisolver is not an LLM based PDE solver.**
>
> You commented in your summary of our paper that **“In this paper, they add symbolic representation of the PDE operator to an LLM based PDE solver.”**
>
> In fact, we don't propose an LLM based PDE solver. We employ an LLM as a frozen encoder to get the symbolic embedding of a PDE. The frozen embedding is then fed into our model to inform the symbolic information of the PDE. The coefficients, boundary conditions, external forces and geometry information are all represented as scalars or fields and encoded by our proposed Universal Component Embedding approach. Thus, the main body of Unisolver is a conditional Transformer, where LLM just performs as an embedding module. To make this issue clear, we have revised the paper by updating the model architecture in $\underline{\text{Figure 3 of the revised paper}}$.
>
> **(2) About Topic: Neural PDE solvers hold significant advantage in efficiency than traditional numerical methods.**
>
> You commented that **"PDE solvers are a machine learning approach to a problem where, if the inputs are properly specified, and the PDEs is well-posed, there is no need for machine learning"**.
>
> We respectfully disagree with your opinion. In comparison to traditional solvers, Neural PDE solvers have shown great advantages in both efficiency and generalizability. For example, FNO can "learn an entire family of PDEs, in contrast to classical methods which solve one instance of the equation" [1]. After training on a wide range of PDEs, Unisolver is able to generalize to PDEs with varying PDE components. Although the training time of Unisolver is relatively long, we can generate the solution on unseen PDEs much faster than traditional numerical PDE solvers. The inference efficiency benefit should not be underestimated. Please think about GPT-4, the training period can be very long, while once the model is well trained it can support very fast and parallel inference.
>
> Moreover, we provide the inference times and memory consumption for each model to predict a single frame on the HeterNS dataset, along with the calculation time and memory consumption for the numerical solver which is used to generate the dataset to calculate the next frame, as summarized in  $\underline{\text{Appendix H.5 of the revised paper}}$. Unisolver is 1,000+ times faster than the numerical solver.
>
> [1] Fourier Neural Operator for Parametric Partial Differential Equations. ICLR 2020.
>
> **(3) About Related Work: The most widely acknowledged advantage of Neural Operators lies in their efficiency, rather than in their ability to operate without requiring a specific equation.**
>
> You commented that **"This paper critiques existing methods as not taking full advantage of the data, but this is a misrepresentation of the learning problem for previous methods, which are expressly designed to solve the PDE without knowing the form of the equation."**
>
> Actually, instead of the ability to solve PDE without exact equations, the most acknowledged benefit of Neural Operators is their efficiency in solving new tasks. We respectively quote some sentences in the previous papers, such as "As a result, the learning-based method can be orders of magnitude faster than the conventional solvers." in FNO (ICLR 2021) and "Serving as surrogate models, neural operators can be more than five orders of magnitude faster than traditional numerical solvers" in DPOT (ICML 2024). Under this spirit, Unisolver is one step towards making the neural PDE solvers applicable to more diverse scenarios, which is of great value in practice.
>
> Further, all the previous methods cited in this paper (e.g., FNO, DPOT, PDEformer,f and ICON) adopt the simulation data for experiments and pursue an efficient surrogate for numerical methods. Since the equation information is just off the shelf in this context, we believe that Unisolver's setting is natural and reasonable.

---

> ### Author Response · Authors · 2024-11-20
> **Response to Reviewer uAAv [Part 2]**
>
> **(4) About Experiments: Many of our experiments follows previously used benchmarks and metrics, and directly compare with officially reported results of baselines.**
>
> You commented that **"There are not standard datasets with standard performance measures: these are new datasets defined by the authors, and performance measures and baselines defined by the authors."**
>
> We respectfully disagree with your opinion. Firstly, we propose the HeterNS dataset by extending the **standard Navier-Stokes dataset** proposed in FNO with broader viscosity coefficients and external forces. The other two datasets, namely the 1D time-dependent PDEs and the 2D mixed PDEs, are proposed by PDEformer and collected by DPOT respectively. Secondly, the performance measurement metric—**Relative L2** we used is widely recognized within the Neural PDE solver field. This metric is considered standard for assessing model performance, as used by FNO, PDEformer and DPOT. Lastly, we compare Unisolver against the official implementation of each baseline in the HeterNS benchmark. The only change we make to these baselines is to augment these models with PDE components to ensure that **they get the same input as Unisolver.** As for the 1D time-dependent PDEs and 2D mixed PDEs benchmarks, we compare Unisolver against PDEformer and DPOT in a head-to-head manner without making any changes to them.
>
> We hope the above clarifications can help the reviewer understand the value of our work better. Next, we will answer the detailed questions.
>
> > **Q1:** "It was not clear exactly what the PDE was trained on, in for example, Table 2. "  "Inputs, outputs, and data need to be specified."
>
> We have provided detailed descriptions of the PDEs we trained on in $\underline{\text{Appendix E of original submission}}$, including the the analytical form of each equation. Following your suggestion, we have also included a new figure $\underline{\text{Figure 2 of main text}}$ to illustrate the input and output of our model.
>
> > **Q2:** "Also the modiﬁcation to the baselines is not clearly described."  "These are no longer baselines, but new methods."
>
> We appreciate your concerns regarding our comparisons with the baselines. Actually, we have already provided implementation details of each baseline in $\underline{\text{Appendix F of original submission}}$. We would like to clarify that our comparisons are indeed fair. In the HeterNS benchmark, we input external force and coefficient information in **numerical form** **rather than as text** to Unisolver. For consistency, we chose to concatenate this information with the baseline input, allowing for a meaningful and fair comparison.
>
> Additionally, we run FNO without concatenating the PDE components and report these results below, demonstrating that FNO’s performance declines without access to this information. That is why we "augment these baselines" in our original submission instead of directly using their original design.
>
> | Viscosity $\nu $       | 1e-5   | 5e-5   | 1e-4   | 5e-4    | 1e-3   | 8e-6   | 3e-5   | 8e-5   | 3e-4   | 8e-4   | 2e-3   |
> | ---------------------- | ------ | ------ | ------ | ------- | ------ | ------ | ------ | ------ | ------ | ------ | ------ |
> | FNO without components | 0.0701 | 0.0259 | 0.0168 | 0.00395 | 0.0017 | 0.0859 | 0.0408 | 0.0163 | 0.0106 | 0.0090 | 0.1876 |
> | FNO with components    | 0.0669 | 0.0225 | 0.0114 | 0.0031  | 0.0011 | 0.0702 | 0.0373 | 0.0141 | 0.0088 | 0.0084 | 0.2057 |
>
> | Force $\omega$         | 1      | 2      | 3      | 0.5    | 1.5    | 2.5    | 3.5    |
> | ---------------------- | ------ | ------ | ------ | ------ | ------ | ------ | ------ |
> | FNO without components | 0.0928 | 0.0960 | 0.2769 | 0.6034 | 0.4962 | 0.4320 | 0.8194 |
> | FNO with components    | 0.0640 | 0.0661 | 0.1623 | 1.1100 | 0.1742 | 0.1449 | 0.2974 |
>
> > **Q3:** "Motivated by the mathematical structure of PDEs, we define the concept of complete PDE components, classify them into domain-wise and point-wise categories, and derive a decoupled conditioning mechanism for introducing physics information into PDE solving. This is a concept completely original to the paper, which ignores existing expertise on PDE operators, and is not clearly explained. It's also not clear how it corresponds to what they do in the paper."
>
> Firstly, We would like to clarify that **the concept of “Complete PDE Component” is specific to conditional deep neural networks and is not intended to serve as a formal mathematical theorem.** In the context of deep learning, it is widely recognized that providing models with fine-grained control conditions can lead to more deterministic and desired results (See [1,2]). Our goal here is to define the conditions necessary for a Neural PDE solver to function effectively.

---

> ### Author Response · Authors · 2024-11-20
> **Response to Reviewer uAAv [Part 3]**
>
> Additionally, we categorize these PDE components into domain-wise and point-wise components **also in the context of deep learning**. Domain-wise components such as coefficients and boundary condition types are generally represented scalars or fixed-dimensional vectors that are uniform across the domain, and point-wise components are generally functions corresponding to specific spatial and temporal coordinates. To integrate these components effectively, we design tailored embedding strategies that incorporate them as deep conditions in Unisolver.
>
> We have highlighted the deep learning context in the $\underline{\text{revised paper}}$.
>
> [1] Adding Conditional Control to Text-to-Image Diffusion Models, ICCV 2023
>
> [2] Scalable Diffusion Models with Transformers, ICCV 2023
>
> > **Q4:** "Theorem 1: This is not a theorem." "The solution expressed in the theorem is particular to the equation, and there do not exist similar explicit solutions for most PDEs."
>
> Thanks for your rigorous review. We acknowledge that this particular PDE simply serves as a motivating example to show the concept of PDE components. We categorize these PDE components in the context of deep learning. The model architecture is motivated by the categorization and also builds upon empirical experiments. We have modified this part in the $\underline{\text{revised paper}}$.
>
> > **Q5:** "All the equation prompts had constant coefficients. How would you prompt a two dimensional equation, with variable coefficients?"
>
> As stated in $\underline{\text{Equation formulation paragraph in Line 233 of the original submission}}$, we don't use an LLM to encode numerical information such as coefficients. The LLM is only responsible for embedding the symbolic information of the PDE. For your further information, it may be hard for LLM to directly solve a PDE. What we want to do in Unisolver is to utilize LLM's basic knowledge in understanding PDEs (Please see visualization in $\underline{\text{the end of page 9}}$).
>
> For a two dimensional equation with variable coefficients, we would use patch embedding to embed the coefficients into deep conditions.
>
> > **Q6:** "How do you distinguish between the exact PDE and incomplete knowledge of the PDE?"
>
> Again, As stated in $\underline{\text{Equation formulation paragraph}}$, we would like to clarify that our LLM embeddings only encodes the Latex code of the PDE, which does not include the numerical part of the PDE. This means that not all PDE information is captured within these symbols. We believe that LLMs, while capable of general mathematical understanding, cannot address all aspects of mathematical problems comprehensively and accurately. The embeddings primarily capture a general understanding relevant to a class of PDEs.
>
> To address incomplete knowledge scenarios, we conduct experiments with “incomplete” PDE information in "Incomplete component scenario" section $\underline{\text{in Line 496 of original submission}}$, where certain components were omitted and replaced with a learnable token to represent the unknown PDE component. These experiments demonstrate that Unisolver still performs well without complete information of the PDE.
>
> > **Q7:** "There is no universal PDE solver, since PDEs do not universally have solutions. Some PDE operators are well-posed, which means there existing unique solutions which are stable under certain types of perturbations of the data."
>
> Thanks for your valuable question. We have revised our title to **Universal Neural PDE Solver** to moderate our claim. In the context of deep learning, "Universal" means that the deep model can solve more diverse distribution (Please see [1]).
>
> Besides, it is important to clarify that our model is not intended to solve every possible PDE. Equations that are not well-posed cannot be solved without sufficient training data. However, our paper focuses on one of the most prominent and cutting-edge problems in the community: whether a trained Neural PDE solver can generalize to more diverse PDEs, which is also the focus of PDEformer and DPOT. To further illustrate our approach, we have modified $\underline{\text{Figure 2,3 in the revised paper}}$ to highlight our definition of "universal" in this context.
>
> [1] Universal Physics Transformers: A Framework For Efficiently Scaling Neural Operators, NeurIPS 2024

---

> ### Author Response · Authors · 2024-11-20
> **Response to Reviewer uAAv [Part 4]**
>
> > **Q8:** "This is an imprecise characterization. There are multiple formulations of PDE problems: these include: (i) learning a solution operator, based on a dataset of solutions on grids, and (ii) learning a single PDE solution, based on incomplete data, as well as knowledge of the PDEs, as well as (iii) learning the PDE operator from sample(s) of (possibly incomplete data."
>
> The categorization of Neural PDE solvers into pure data-driven methods and physics-informed methods is well accepted [1] [2]. Data-driven methods utilize data from numerical solvers or existing experiments, while physics-informed methods only depend on the PDE constraints. A representative example of data-driven methods is the class of neural operators, while a representative example of physics-informed methods is the class of PINNs.
>
> It would be highly appreciated if you provided references that support your mentioned specific categorization.
>
> [1] Physics-informed neural operator for learning partial differential equations. ACM/JMS Journal of Data Science 2024
>
> [2] Transolver: A fast transformer solver for pdes on general geometries. ICML 2024
>
> > **Q9:** "What exactly is the training data, expressed in terms of the solutions to which PDEs? Was it all the PDE solutions, were the text prompts included with the data? What is the test data? An initial condition? Along with the text of the PDE? Please describe what were the inputs to the training dataset, what were the inputs to the model (besides the latex text prompt for the PDE) and the models trained on a dataset of PDEs for each of the three main cases?"
>
> We have provided detailed descriptions of training data in $\underline{\text{Appendix E of the original submission}}$. To be more clear, we have also included or modified figures in the main text ($\underline{\text{Figure 2,3 in revised paper}}$) to illustrate the input and output of our model more clearly.
>
> > **Q10:** "what is meant by "favorable generalizability"?"
>
> Favorable generalizability means the **empirical improvement** on the standard benchmarks we considered, where our model significantly outperforms numerous baseline models in a zero-shot generalization setting. For detailed information on the benchmark design and baseline implementations, please refer to $\underline{\text{Appendix E and F of the original submission}}$.
>
> >**Q11:** "This is not clear: what are 'point-wise deep conditions'"
>
> As stated in $\underline{\text{Sec 3.2 of the original submission}}$, we provide a detailed explanation of how to embed all considered PDE components into deep PDE conditions. The point-wise deep conditions are obtained by embedding each point-wise component and then aggregating them, as illustrated in $\underline{\text{Figure 3 of the original submission}}$, which is relatively clear in the context of deep learning.
>
>
>
> **Again, we have to point out that Unisolver is not an LLM based solver**. The LLM serves as a frozen encoder to encode the symbolic information of the PDE, while the numerical part of the PDE are encoded either by an MLP layer or a Patch-Embedding module.

---

> ### Author Response · Authors · 2024-11-25
> **Request of Reviewer’s attention and feedback**
>
> Dear Reviewer uAAv,
>
> We kindly remind you that there are only **two days left** of the two-week Reviewer-author discussion period. So please kindly let us know if our response has addressed your concerns. We will be happy to deal with any further issues/questions.
>
> We respectfully clarify some misunderstandings about our work:
>
> - We clarify that **Unisolver is not an LLM-based PDE solver**, as the LLM serves only as a frozen encoder for symbolic embedding.
> - We emphasize the **efficiency and generalizability advantages** of neural PDE solvers over traditional numerical methods.
> - We highlight that in recent papers, the most widely acknowledged advantage of Neural Operators lies in their efficiency, rather than in their ability to operate without requiring a specific equation.
> - We clarify that the datasets and performance metrics we use are all standard and that we make fair comparisons with baselines.
>
> We also made every effort to address the concerns you suggested:
>
> - We have revised our title to **Universal Neural PDE Solver** to moderate our claim.
> - We include a new figure  $\underline{\text{Figure 2}}$  and revise $\underline{\text{Figure 3 of main text}}$​ to better illustrate the problem setup of our work and the input and output of Unisolver.
> - We emphasize that the concept of "Complete PDE Component" is specific to conditional deep neural networks and the categorization of domain-wise and point-wise components is also in the context of deep learning.
> - We have revised $\underline{\text{Section 3.1}}$ to use the vibrating string equation solely as a motivating example for introducing the concepts of domain-wise and point-wise PDE components.
>
> These updates have already been incorporated into the $\underline{\text{revised paper}}$. We hope our responses and revisions have addressed all your concerns effectively.
>
> Thanks again for your dedication to reviewing our paper. Looking forward to your reply.

---

> > ### Comment · Reviewer_uAAv · 2024-11-25
> >
> > I am keeping my rating.

---

> > > ### Author Response · Authors · 2024-11-26
> > > **Respectfully Inquiry the Reason for Your Response**
> > >
> > > Dear reviewer,
> > >
> > > Thank you for the taking the time to review our paper. We believe there have been some misunderstandings in your previous review, and we have made every effort to address them thoroughly in our response. Note that you have decided to keep your rating, we would like to respectfully ask if our response has clarified these misunderstandings or if there are any remaining concerns we can further address.
> > >
> > > Thank you very much for your time, and we look forward to your response.

---

> > > ### Author Response · Authors · 2024-11-27
> > > **Looking forward to your reply. Hope you can detail the reason for your last response.**
> > >
> > > Dear Reviewer uAAv,
> > >
> > > Sincerely thanks for your dedication to our paper. We truly appreciate the care and thoughtfulness you have invested in such a detailed review. Your remarks on "There is no universal PDE solver" were particularly valuable, which made us revise the title to **Universal Neural PDE Solver** for scientific rigor.
> > >
> > > Following your suggestions, we have discussed all your mentioned weaknesses in detail, including clarifications on the problem setup, clarifications on baselines, and the representation of the motivating example. **All the updates have been included in our $\underline{\text{revised paper}}$**. Meanwhile, we respectfully point out that **there may have been some misunderstandings in your previous review**, and we have included a detailed clarification in our previous rebuttal. Hope it can resolve your concerns about the value of our paper.
> > >
> > > Since in your last response, you did not provide the reason for your "keep rating" decision, we kindly remind you that: as highlighted in the **ICLR Reviewer Guidelines** (https://iclr.cc/Conferences/2024/ReviewerGuide), the final recommendation requests the reviewer "**state your reasoning and what did/didn’t change your recommendation throughout the discussion phase.**" In this spirit, we sincerely hope you can clearly indicate whether our rebuttal has addressed your concerns. We highly value your expertise and believe your insightful comments will be instrumental in refining this paper further.
> > >
> > > We believe that this paper (**more than 30 pages** with extensive experiments, ablations and visualizations as well as **inference code**) can be a good supplement to current research on "generalizable Neural PDE Solver".
> > >
> > > Thanks again for your thoughtful review. We eagerly await your reply and are happy to answer any further questions.
> > >
> > > All the best,
> > >
> > > Authors

---

> > > ### Author Response · Authors · 2024-11-28
> > > **We are anticipating your feedback**
> > >
> > > Dear Reviewer uAAv,
> > >
> > > Thank you once again for your valuable and constructive review, which has greatly inspired us to improve our paper further.
> > >
> > > We eagerly await your reply and are happy to answer any further questions. We kindly remind you that **the deadline for uploading the revised PDF is November 27th, just a few hours away.** After that, we may not have the opportunity to incorporate any further revisions based on your feedback.
> > >
> > > Sincere thanks for your dedication!
> > >
> > > Authors

---

> > > > ### Comment · Reviewer_uAAv · 2024-11-28
> > > >
> > > > I read over my original review and the responses.
> > > >
> > > > PDE solvers are a machine learning approach to a problem where, if the inputs are properly specified, and the PDEs is well-posed, there is no need for machine learning: standard scientific computing results give very precise and well-understood solvers. Machine learning approaches need to be properly motivated, and the problem needs to be carefully defined.
> > > >
> > > > In this paper, the problem is not clearly defined. The authors "employ an LLM as a frozen encoder to get the symbolic embedding of a PDE. The frozen embedding is then fed into our model to inform the symbolic information of the PDE. The coefficients, boundary conditions, external forces and geometry information are all represented as scalars or fields and encoded by our proposed Universal Component Embedding approach. Thus, the main body of Unisolver is a conditional Transformer, where LLM just performs as an embedding module."
> > > >
> > > > From the point of view of a downstream application, or of a researcher trying to add to the body of useful work on PDE solvers, I cannot recommend acceptance of this paper.

---

> > > > > ### Author Response · Authors · 2024-11-28
> > > > > **Thanks for your response and more explanations about your concerns**
> > > > >
> > > > > Dear Reviewer uAAv,
> > > > >
> > > > > Thank you for your follow-up comment. We greatly appreciate your time and effort in reviewing our responses. We would like to address the points raised in your feedback in detail below:
> > > > >
> > > > > > **Q1: "PDE solvers are a machine learning approach to a problem where, if the inputs are properly specified, and the PDE is well-posed, there is no need for machine learning."**
> > > > >
> > > > > As clarified in $\underline{\text{Misunderstanding (2) of our response in Part 1}}$, we would like to emphasize that, compared to traditional solvers, neural PDE solvers offer significant advantages in both **efficiency and generalizability**, even when **"the inputs ... well posed"**. We have provided an efficiency comparison between neural operators and numerical solvers in $\underline{\text{Appendix H.5 of the revised paper}}$, demonstrating that Unisolver is over 1,000 times faster than traditional numerical solvers. For your convenience, we also present this comparison again here, which shows the inference (calculation) time and memory consumption for each model and numerical solver to predict a single frame on the HeterNS dataset.
> > > > >
> > > > > |                  | Average Memory Usage/MB | Average Inference (Calculation) Time/s |
> > > > > | ---------------- | ----------------------- | -------------------------------------- |
> > > > > | FNO              | 730                     | 0.0042                                 |
> > > > > | Unisolver        | 554                     | 0.0054                                 |
> > > > > | Numerical Solver | 524                     | 7.26                                   |
> > > > >
> > > > > Moreover, Physics-informed Neural Networks (PINNs) are also a prominent machine learning approach built upon complete PDE information [1] [2]. If you believe that "if the inputs are properly specified ... machine learning," **we would like to respectfully ask whether you also consider PINNs to be unnecessary**. In comparison to PINNs, Unisolver combines the efficiency advantages of data-driven methods with the usage of PDE information, allowing Unisolver to generalize effectively across diverse PDEs while maintaining computational efficiency.
> > > > >
> > > > > Based on these points, **we believe that even when "the inputs ... well posed", machine learning approaches are still useful**. We kindly hope that you could reconsider the value of the problem addressed in this paper.
> > > > >
> > > > > [1] Raissi et al. Physics-informed neural networks: A deep learning framework for solving forward and inverse problems involving nonlinear partial differential equations. Journal of Computational physics, 2019.
> > > > >
> > > > > [2] Karniadakis et al. Physics-informed machine learning. Nature Reviews Physics, 2021
> > > > >
> > > > > > **Q2: "In this paper, the problem is not clearly defined.  The authors "employ an ... an embedding module."**
> > > > >
> > > > > We have to respectfully point out that your comment **cites the description of our method, not our problem definition**. In response to your concern in your original review that "In this paper, the problem is not clearly defined", we have added a new figure, $\underline{\text{Figure 2 in the revised paper}}$ to illustrate our problem setup. Additionally, the problem setup paragraph has been updated for clarity. Specifically, we define the **universal neural PDE solving task** as to approximate the map from initial observations and PDE components to output physical quantities on a given mesh across a diverse training dataset and generalize in a flash to unseen PDEs. We kindly request you to read $\underline{\text{the problem setup paragraph in Section 3 of the revised paper}}$ for details. In summary, **we believe our clarification for the problem definition would address your concerns.**
> > > > >
> > > > > > **Q3: "From the point of view of a downstream application, or of a researcher trying to add to the body of useful work on PDE solvers, I cannot recommend acceptance of this paper."**
> > > > >
> > > > > Unisolver can become an efficient surrogate model for numerical PDE solvers and produce a solution 1,000 times faster. For instance, **speed is critical for industrial applications**, and neural PDE solvers like Unisolver offer a compelling solution. Recently, **NVIDIA** has introduced **Omniverse Blueprints**, a platform that leverages deep learning for PDE simulations, enabling real-time creation and interaction with digital twins (https://developer.nvidia.com/blog/rapidly-create-real-time-physics-digital-twins-with-nvidia-omniverse-blueprints/). This is done by training AI models on simulation data where **complete PDE information is known and PDEs are well-posed**. These tools demonstrate that **even when complete PDE information is available**, machine learning solutions can effectively address the critical demands for **real-time and frequent simulations**, highlighting their practicality and potential in industrial environments.
> > > > >
> > > > > We hope these clarifications address your concerns effectively and we are happy to answer any further questions.
> > > > >
> > > > > Sincerely,
> > > > >
> > > > > Authors

---

> > > > > ### Author Response · Authors · 2024-11-29
> > > > > **Further Clarification of Problem Definition**
> > > > >
> > > > > Dear reviewer uAAv,
> > > > >
> > > > > To further enhance your understanding of our experimental setups, we summarize the **inputs and outputs of Unisolver** on each benchmark here. Note that detailed descriptions about the benchmarks and implementation details of baselines and our method have been presented in $\underline{\text{Appendix E and F of the original submission}}$, which also include the details about our **training and testing setup**. We also need to highlight that the **problem setup** of our paper has been completely illustrated in $\underline{\text{Figure 2 and the problem setup paragraph in Section 3 in the revised paper}}$.
> > > > >
> > > > > 1. **The HeterNS benchmark ($\underline{\text{Appendix E.1 of the original submission}}$):**
> > > > >
> > > > >    The HeterNS is an extension of the NS benchmark proposed by FNO, where the only varied PDE components are viscosity coefficients and external force terms.
> > > > >
> > > > >    - **Inputs:** Ten previous vorticity frames and the varied PDE components (viscosity coefficients and external force terms).
> > > > >    - **Outputs:** The next consecutive vorticity frame.
> > > > >
> > > > > 2. **The 1D time-dependent PDEs benchmark:  ($\underline{\text{Appendix E.2 of the original submission}}$):**
> > > > >
> > > > >    - **Inputs:** The initial conditions and the varied PDE components (equation symbols, coefficients, forces and boundary conditions).
> > > > >    - **Outputs:** The spatial-temporal solution field.
> > > > >
> > > > > 3. **The 2D mixed PDEs benchmark:  ($\underline{\text{Appendix E.3 of the original submission}}$):**
> > > > >
> > > > >    - **Inputs:** Ten previous solution frames and the varied PDE components (equation symbols, coefficients, forces, geometry and boundary conditions).
> > > > >    - **Outputs:** The next consecutive solution frame.
> > > > >
> > > > > Thank you for your time and we eagerly await your reply.
> > > > >
> > > > > Best regards,
> > > > >
> > > > > Authors

---

> ### Author Response · Authors · 2024-12-01
> **Discussion Period Ending Soon**
>
> Dear reviewer uAAv,
>
> Thanks for your time in reviewing our paper.
>
> As the discussion period is nearing its end, we are eager to know whether our response has addressed your concerns about the **problem setup** and the value of machine learning for "properly specified inputs and well-posed PDEs."
>
> We sincerely look forward to your feedback.
>
> Best regards,
>
> Authors

---

> ### Author Response · Authors · 2024-12-03
> **Eagerly awaiting your feedback**
>
> Dear reviewer uAAv,
>
> We sincerely thank you once again for your time reviewing our work.
>
> Since there are only 9 hours left before the deadline for the reviewer-author discussion, we would greatly appreciate your feedback on whether our responses have adequately addressed your concerns.
>
> For clarity, we highlight some key points from our response here:
>
> - **Our topic is meaningful and practical, which is also actively explored by NVIDIA.** Even when "the inputs ... well-posed", Neural PDE Solvers hold unique advantages in efficiency. Specifically, **Unisolver is over 1,000 times faster than traditional numerical solvers, see our previous response**. This is why some top companies, such as NVIDIA Omniverse Blueprints, still work on Neural PDE Solvers for pure simulation scenarios with complete PDE information. Thus, we respectfully disagree with your opinion that "if the inputs are properly specified, and the PDE is well-posed, there is no need for machine learning".
>
> - **The problem setup is clear in our paper with all the input-output well defined.** Our problem definition has been clearly presented in $\underline{\text{Figure 2 and the problem setup paragraph in Section 3 in the revised paper}}$. Besides, our paper also includes all the inputs and outputs of each benchmark. Thus, we think the problem setup is crystal clear. We do hope you can refer to $\underline{\text{Page 3-4}}$ of our revised paper.
>
> Based on the previous discussion, we think there are some foundational misunderstandings in your review. Since it is nearing the end of the rebuttal period, we do hope you can carefully review our clarification and reconsider your score.
>
> Best regards,
>
> Authors

---

### Author Response · Authors · 2024-11-20
**Summary of Revisions [Part 2]**

The reviewers also raised insightful and constructive concerns. We made every effort to address all the concerns by providing detailed clarification and requested experiments. Here is the summary of the major revisions:

1. **Clarify the problem setup and add a new figure to better illustrate the concept of universal PDE solving. (Reviewer uAAv, awFj):** We revise the problem setup paragraph in $\underline{\text{Section 3}}$ and add $\underline{\text{Figure 2}}$ to better illustrate our problem setup and the concept of universal PDE solving. We define our task of universal PDE solving in the deep learning context as to approximate the input-PDE-output mapping across a diverse training dataset and generalize in a flash to unseen PDEs. Furthermore, we **revise our title** to "Unisolver: PDE-Conditional Transformers Are **Universal Neural PDE Solvers**" to restrict our claim within the neural PDE solver domain.
1. **Revise Section 3.1 and use the vibrating string equation merely as a motivating example. (Reviewer uAAv, awFj):** We have revised $\underline{\text{Section 3.1}}$ to use the vibrating string equation solely as a motivating example for introducing the concepts of domain-wise and point-wise PDE components. We also remove the proof in Appendix E for compactness. We further emphasize that the categorization of PDE components is in the context of deep learning, and that our model architecture design is clearly motivated from this categorization and also based on empirical experiments.
1. **Clarify one ablation and provide three more ablations on PDE symbol embeddings. (Reviewer 1Nnh, fFn6):** First, we clarify that the ablation with "random vector" in $\underline{\text{Section 4.4 of original submission}}$ actually employs **"orthogonal random vectors"**. For new ablations, the first one uses LLM to encode the number of non-zero terms in the PDE and the second one uses LLM to embed "wrong" PDE formulations. And the third one manually constructs a informative representation and does not use LLM to embed it. These ablations further show the benefit of using LLM embeddings. See $\underline{\text{Appendix B in the revised paper}}$ for details.
1. **Discuss the advantages of data-driven Neural PDE solvers with respect to numerical methods and PINNs. (Reviewer uAAv, awFj):** We emphasize that data-driven Neural PDE solvers can achieve significant efficiency gains and generalizability compared to numerical methods and PINNs, and can generate solutions on new PDEs in a flash without retraining.
1. **Discuss the inference efficiency and computational cost. (Reviewer uAAv, 1Nnh, awFj):** We provide an inference time and memory usage comparison on the HeterNS benchmark. The computational efficiency of the numerical method which is used to generate HeterNS to calculate the next frame,  is reported as well. Unisolver is relatively efficient and is approximately 1,000 times faster than the numerical solver. Detailed comparison can be found in $\underline{\text{Appendix H.5 in the revised paper}}$.
1. **Provide experiments about the effect of joint training and long trajectory prediction.** **(Reviewer awFj, fFn6):** We further verify the advantage of joint training. It significantly boosts model's performance even though these types of PDEs do not overlap. We also provide an additional experiment on longer trajectory prediction, and Unisolver demonstrates strong zero-shot performance. These results can be found in $\underline{\text{Appendix C.1, H.3 in the revised paper}}$.
1. **Release the inference code along with the checkpoint for 1D time-dependent PDEs.**  We have provided an example **inference code for the 1D time-dependent PDEs** benchmark in the supplementary material, along with the **checkpoint of our largest Unisolver model**, which was trained for **3000 A100 hours on 3 million samples**. We promise to also public both training and inference codes upon acceptance.

All the revisions have been included in the $\underline{\text{revised paper highlighted in blue}}$. The valuable suggestions from reviewers are very helpful for us to revise the paper to a better shape. We'd be very happy to answer any further questions.

Looking forward to the reviewer's feedback.

---

### Author Response · Authors · 2024-11-20
**Summary of Revisions [Part 1]**

We sincerely thank all the reviewers for their insightful reviews and valuable comments, which are instructive for us to improve our paper further.

This paper presents the Unisolver to solve a wide scope of PDEs simultaneously. Going beyond prior purely data-driven neural PDE solvers, we define a complete set of PDE components, correspondingly embed them as domain-wise (e.g. equation symbols) and point-wise (e.g. boundaries) conditions and utilize a conditional Transformer to adaptively fuse them with initial observations. **Experimentally, Unisolver achieves consistent state-of-the-art results on three challenging large-scale benchmarks, showing impressive gains and endowing favorable generalizability and scalability.**

The reviewers generally hold positive opinions towards our paper, in that the proposed method tackles **"challenging tasks"** with a **"simple but powerful architectural design"** and a **"novel and decoupled conditioning mechanism for introducing physics information into PDE solving"**. They find our approach **"novel and clearly described"** with **"reasonable motivation and thorough analysis"**, and our paper is **"clearly written and easy to follow"**. They also praise our **"good experimental design"** and that our architecture shows **"stronger performance"**. We have provided **"substantial implementation details"**, and the ablation study **"validates the importance of the proposed ideas"**.

---

### Meta-Review · Area_Chair_T7oP · 2024-12-20

**Metareview:**

The manuscript proposes an approach of pre-training for solving PDEs based on text prompts. While the idea is interesting, the significance of this approach to scientific computing is unclear; and as pointed out by the expert reviewers, the presentation is rather unclear and seems to lack some basic understanding of numerical solutions to PDEs. While the reviewers opinions are quite disparate, after carefully reading the manuscript and the discussion, the metareviewer agrees with the negative reviews and cannot recommend this manuscript.

**Additional Comments On Reviewer Discussion:**

While the authors are quite engaging in the discussion phase, from the meta-reviewer's read, their response does not answer the main concerns and criticism from the reviewers, in particular, those from reviewer uAAv.

---

### Decision · Program_Chairs · 2025-01-22

Reject